# Competitive Advantage Attacks to Decentralized Federated Learning

**Yuqi Jia**[1]    **Minghong Fang**[2]    **Neil Gong**[1]
[1]Duke University,    [2]University of Louisville
[1]{yuqi.jia, neil.gong}@duke.edu,    [2]minghong.fang@louisville.edu

## Abstract

Decentralized federated learning (DFL) enables clients (e.g., hospitals and banks) to jointly train machine learning models without a central orchestration server. In each global training round, each client trains a local model on its own training data and then they exchange local models for aggregation. In this work, we propose SelfishAttack, a new family of attacks to DFL. In SelfishAttack, a set of selfish clients aim to achieve *competitive advantages* over the remaining non-selfish ones, i.e., the final learnt local models of the selfish clients are more accurate than those of the non-selfish ones. Towards this goal, the selfish clients send carefully crafted local models to each remaining non-selfish one in each global training round. We formulate finding such local models as an optimization problem and propose methods to solve it when DFL uses different aggregation rules. Theoretically, we show that our methods find the optimal solutions to the optimization problem. Empirically, we show that SelfishAttack successfully increases the accuracy gap (i.e., competitive advantage) between the final learnt local models of selfish clients and those of non-selfish ones. Moreover, SelfishAttack achieves larger accuracy gaps than poisoning attacks when extended to increase competitive advantages. Our code and data are available at: https://github.com/jyqhahah/SelfishAttack.

## 1   Introduction

In *decentralized federated learning (DFL)* [8, 15, 17, 33, 9, 34, 29, 35], a group of clients (e.g., hospitals or banks) collaboratively train machine learning models without relying on a central server. Each client maintains a *local model* (called *pre-aggregation local model*), trains it using private data, exchanges a *shared model* with others, and aggregates shared models from all clients as a new local model (called *post-aggregation local model*) through an *aggregation rule*, e.g., FedAvg [24]. In non-adversarial settings, all clients converge to identical models after each round. Compared to centralized federated learning (CFL), which requires a trusted coordinator, DFL offers enhanced robustness by eliminating single points of failure and is better suited to scenarios, e.g., the clients are hospitals or banks, where clients cannot agree on a central authority.

This paper reveals a new vulnerability in DFL: its decentralized nature opens the door to a unique class of insider attacks, which we call *SelfishAttack*. In this attack, a subset of *selfish clients* collude to compromise the training process, aiming to: 1). improve their final local models beyond what they could achieve by training among themselves (called **utility goal**), and 2). outperform the remaining *non-selfish clients* (called **competitive-advantage goal**).

Unlike conventional poisoning attacks [3, 6, 10, 31, 13], which assume external adversaries that compromise or inject malicious clients to cause indiscriminate misclassification, SelfishAttack is an internal, collusion-based threat. The selfish clients participate fully in the DFL process but selectively manipulate only the models shared with non-selfish participants. Their goal is not to poison

39th Conference on Neural Information Processing Systems (NeurIPS 2025).

models outright, but to widen the performance gap in their favor—while still benefiting from collaboration. This makes SelfishAttack more subtle and targeted than traditional attacks.

For instance, in medical DFL applications such as Alzheimer's diagnosis from 3D brain MRI [28, 32], several hospitals may collaborate to train diagnostic models. However, a few self-interested centers could covertly apply SelfishAttack to gain diagnostic advantages, improving their models by leveraging external updates while degrading those of non-collaborators—all without disrupting the overall training. This scenario underscores the risks of collusion in decentralized systems, where no central authority exists to detect or prevent such subtle manipulations.

To formalize the attack, we cast SelfishAttack as an optimization problem balancing two loss terms corresponding to its dual objectives. The key idea is to craft the shared models sent to non-selfish clients such that: 1). their post-aggregation local models remain close to their pre-aggregation models (ensuring they continue to contribute useful updates), while 2). being strategically perturbed to degrade their model performance relative to the selfish group. For aggregation rules such as FedAvg, Median, and Trimmed-mean, we derive the closed-form expressions for the optimal shared models. For more complex rules (e.g., Krum [3], FLTrust [5], FLDetector [37], RFA [27], FLAME [25]), we adapt these expressions to construct effective attacks.

We evaluate SelfishAttack on three diverse datasets, e.g., CIFAR-10, FEMNIST, and Sent140, and compare it against poisoning-based baselines adapted to our setting. Our experiments show that SelfishAttack can simultaneously achieve the utility and competitive-advantage goals. For example, on CIFAR-10 with 30% selfish clients, SelfishAttack improves their final model accuracy by at least 11% over that of the non-selfish clients. Moreover, it consistently outperforms poisoning-based baselines in both absolute model performance and accuracy gap. Our contributions can be summarized as follows:

- We propose SelfishAttack, a new family of attacks for DFL where selfish clients seek competitive advantage and maintain local model utility.

- We theoretically derive optimal shared models of selish clients that balance these goals under three aggregation rules, including FedAvg, Median, and Trimmed-mean.

- Experiments on three benchmark datasets show that SelfishAttack effectively achieves both attack goals and outperforms existing poisoning attacks.

## 2 Preliminaries and Related Work

### 2.1 Decentralized Federated Learning

In a Decentralized Federated Learning (DFL) system, $N$ clients collaboratively train a model without a central server. Each client $h$ has its own dataset $\mathcal{D}_h$ and maintains a local model. The training process proceeds in *global training rounds*, where each round includes three steps:

**Step I. Local Training:** Each client trains its local model using local data and obtains a *pre-aggregation local model* $\hat{\boldsymbol{w}}_h$.

**Step II. Model Sharing:** : Client $h$ exchanges *shared models* $\boldsymbol{w}_{(h,h')}$ to each other client $h'$. In honest settings, shared models are identical to pre-aggregation local models.

**Step III. Model Aggregation:** : Each client aggregates received shared models using an *aggregation rule* to update its *post-aggregation local model* $\boldsymbol{w}_h$.

The process repeats for multiple rounds until convergence. In non-adversarial settings, all clients typically converge to the same local model. See Appendix A.1 for more details.

### 2.2 Poisoning Attacks in FL

Poisoning attacks in FL aim to degrade the performance of global or local models and can be broadly categorized into untargeted and targeted attacks. Existing works such as Gaussian attack [3] and Trim attack [10] show that adversaries can manipulate shared models to corrupt learning. We provide further background and details on poisoning attack strategies in Appendix A.2.

## 2.3 Robust Aggregation Rules

To defend against adversarial behaviors, several Byzantine-robust aggregation rules have been proposed, including Median [36], Trimmed-mean [36], and Krum [3], which aim to limit the influence of outlier models. More advanced defenses [5, 27, 11, 25, 14, 15, 7, 30, 19, 37, 12] like FLTrust [5], FLDetector [37], RFA [27], and FLAME [25] use additional mechanisms such as trust scoring or clustering. We summarize these defenses in Appendix A.3.

## 3 Threat Model

**Attacker's goal:** We consider a colluding subset of clients, termed *selfish clients*, as the attacker. Their goal is to compromise the DFL training to achieve two objectives: 1). The *utility goal*: selfish clients aim to obtain more accurate local models than those learned by training solely among themselves. Formally, let $M_1$ be the local model obtained when selfish clients train solely among themselves, and $M_2$ the model learned when they participate in full DFL. The utility goal requires that $M_2$ is more accurate than $M_1$. 2). The *competitive-advantage goal*: the selfish clients aim to obtain more accurate local models than those of the non-selfish clients, gaining practical benefits such as improved service quality or increased revenue in competitive domains.

**Attacker's capabilities:** A selfish client sends its true pre-aggregation local model to other selfish clients but can arbitrarily manipulate the models shared with non-selfish clients. In contrast, non-selfish clients follow standard DFL behavior and share their pre-aggregation local models with all peers. We assume the standard resilience condition $N \geq 3m + 1$ [21], where $m$ is the number of selfish clients and $N$ is the total number of clients. Fig.1 illustrates the attack mechanism.

**Attacker's knowledge:** As legitimate participants, selfish clients know the aggregation rule and receive the pre-aggregation local models from all non-selfish clients. They also share their own pre-aggregation local models among themselves.

**Comparison with poisoning attacks:** Unlike traditional poisoning attacks [3, 6, 10], which typically involve external adversaries injecting faulty updates to indiscriminately degrade model performance, SelfishAttack is an internal, coordinated attack. Poisoning attacks often reduce the performance of all involved clients, including compromised ones, yielding no competitive advantage. In contrast, SelfishAttack carefully manipulates only selected updates to degrade non-selfish clients' models while maintaining or improving the accuracy of selfish clients' own models. As shown in our experiments, compared to poisoning attacks, SelfishAttack effectively degrades non-selfish clients' performance under robust aggregation, while also giving selfish clients a competitive advantage.

## 4 Our SelfishAttack

### 4.1 Overview

SelfishAttack is formulated as an optimization problem minimizing a weighted sum of two losses that capture the utility and competitive-advantage goals. The utility loss ensures that non-selfish clients' local models remain close to their pre-aggregation versions, preserving valuable training information. The competitive-advantage loss encourages divergence from the aggregation that would occur without selfish clients, degrading non-selfish clients' models. As the objective is non-differentiable w.r.t. shared models, we instead derive the optimal post-aggregation local model and then construct corresponding shared models that achieve it under aggregation rules such as FedAvg, Median, and Trimmed-mean. If other aggregation rules are used, we apply shared models optimized for these three as a general strategy. Finally, we propose a criterion for deciding when to start attack. Algorithm 1 in Appendix shows the complete algorithm.

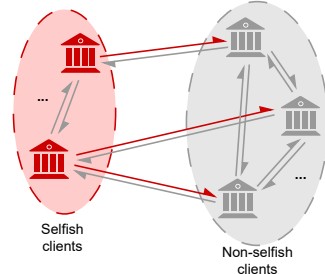

Selfish clients

Non-selfish clients

Figure 1: An illustration of SelfishAttack. Non-selfish clients share their pre-aggregation models with all; selfish clients send tailored models to non-selfish clients to manipulate their updates.

### 4.2 Formulating an Optimization Problem

We assume $n$ non-selfish and $m$ selfish clients in the DFL system, denoted $0, \ldots, n-1$ and $n, \ldots, n+m-1$, respectively. In each round, selfish clients craft shared models to manipulate

the aggregation of a target non-selfish client $i$. Let $\hat{\boldsymbol{w}}_i$ and $\boldsymbol{w}_i$ denote the pre- and post-aggregation local models of $i$, and $\tilde{\boldsymbol{w}}_i$ its model had only non-selfish clients participated. Table 4 in Appendix summarizes our important notations.

To achieve the **utility goal**, selfish clients need to preserve the learning benefits from non-selfish clients. Therefore, they can keep the non-selfish client $i$'s post-aggregation model $\boldsymbol{w}_i$ close to its original local model $\hat{\boldsymbol{w}}_i$. We quantify this as the squared Euclidean distance $\|\boldsymbol{w}_i - \hat{\boldsymbol{w}}_i\|^2$. To achieve the **competitve-advantage goal**, selfish clients should reduce the accuracy of non-selfish clients, so they can drive the non-selfish client $i$'s post-aggregation model away from the model $\tilde{\boldsymbol{w}}_i$ it would get without any selfish participants. This is captured by $-\|\boldsymbol{w}_i - \tilde{\boldsymbol{w}}_i\|^2$. Finally, the objective is:

$$\min_{\{\boldsymbol{w}_{(j,i)}\}_{n \leq j \leq n+m-1}} \|\boldsymbol{w}_i - \hat{\boldsymbol{w}}_i\|^2 - \lambda\|\boldsymbol{w}_i - \tilde{\boldsymbol{w}}_i\|^2, \tag{1}$$

where $\boldsymbol{w}_i = \text{Agg}(\{\boldsymbol{w}_{(h,i)}\}_{0 \leq h \leq n+m-1})$, $\tilde{\boldsymbol{w}}_i = \text{Agg}(\{\boldsymbol{w}_{(h,i)}\}_{0 \leq h \leq n-1})$, $\lambda \geq 0$ is a hyperparameter to achieve a trade-off between the two attack goals, and the variables of the optimization problem are shared models $\{\boldsymbol{w}_{(j,i)}\}_{n \leq j \leq n+m-1}$ sent from selfish clients to non-selfish client $i$.

### 4.3 Solving the Optimization Problem

Since the objective function is not differentiable with respect to shared models $\boldsymbol{w}_{(j,i)}$ for many aggregation rules, we reparameterize the optimization over the post-aggregation model $\boldsymbol{w}_i$ and solve it dimension-wise. For each coordinate $k$, the problem becomes a bounded quadratic program according to $\boldsymbol{w}_i[k]$, where $[k]$ indicates the $k$th dimension:

$$\min_{\underline{\boldsymbol{w}}_i[k] \leq \boldsymbol{w}_i[k] \leq \overline{\boldsymbol{w}}_i[k]} (1-\lambda)\boldsymbol{w}_i[k]^2 - 2(\hat{\boldsymbol{w}}_i[k] - \lambda\tilde{\boldsymbol{w}}_i[k])\boldsymbol{w}_i[k] + (\hat{\boldsymbol{w}}_i[k]^2 - \lambda\tilde{\boldsymbol{w}}_i[k]^2), \tag{2}$$

where $\underline{\boldsymbol{w}}_i[k]$ and $\overline{\boldsymbol{w}}_i[k]$ denote the lower and upper bounds of $\boldsymbol{w}_i[k]$, which will be derived for each aggregation rule. The unconstrained minimizer is $\boldsymbol{p}_i[k] = \frac{\hat{\boldsymbol{w}}_i[k] - \lambda\tilde{\boldsymbol{w}}_i[k]}{1-\lambda}$, and the optimal solution $\boldsymbol{w}_i^*[k]$ is either $\boldsymbol{p}_i[k]$ or clipped to the interval bounds. Formally, we have

$$\boldsymbol{w}_i^*[k] = \begin{cases} \boldsymbol{p}_i[k], & \lambda < 1 \wedge \boldsymbol{p}_i[k] \in [\underline{\boldsymbol{w}}_i[k], \overline{\boldsymbol{w}}_i[k]] \\ \underline{\boldsymbol{w}}_i[k] \text{ or } \overline{\boldsymbol{w}}_i[k], & \text{otherwise} \end{cases} . \tag{3}$$

The details and proof can be found in Appendix B. Then we introduce how to craft selfish clients' shared models they send to each non-selfish client $i$ to achieve the optimal solution $\boldsymbol{w}_i^*[k]$ when using FedAvg, Median, or Trimmed-mean as the aggregation rule.

### 4.4 Attacking FedAvg

**Bound derivation:** FedAvg is not robust, so the selfish clients can make $\boldsymbol{w}_i[k]$ arbitrary. To avoid being easily detected, we define upper and lower bounds as $\overline{\boldsymbol{w}}_i[k] = \max_{0 \leq h \leq n-1} \boldsymbol{w}_{(h,i)}[k]$ and $\underline{\boldsymbol{w}}_i[k] = \min_{0 \leq h \leq n} \boldsymbol{w}_{(h,i)}[k]$.

**Shared model construction:** After computing the optimal target $\boldsymbol{w}_i^*[k]$ via Equation 3, we set $m-1$ selfish clients' values to $\boldsymbol{w}_i^*[k]$, and solve for the last one to ensure the $k$th dimension of the post-aggregation local model of client $i$ becomes $\boldsymbol{w}_i^*[k]$. Therefore, we have:

$$\boldsymbol{w}_{(n+1,i)}[k] = \cdots = \boldsymbol{w}_{(n+m-1,i)}[k] = \boldsymbol{w}_i^*[k], \ \boldsymbol{w}_{(n,i)}[k] = (n+1) \cdot \boldsymbol{w}_i^*[k] - \sum_{h=0}^{n-1} \boldsymbol{w}_{(h,i)}[k]. \tag{4}$$

In Theorem 2 in Appendix, we prove that the shared models crafted using Equation 4 constitute the optimal solution to the optimization problem in Equation 1.

### 4.5 Attacking Median

**Bound derivation:** Since Median discards outliers and $m < N/3$, the aggregated value $\boldsymbol{w}_i[k]$ lies within a bounded range. We sort the $k$th-dimensional shared models from non-selfish clients to $i$ in descending order as $\boldsymbol{q}_{0i}[k] \geq \cdots \geq \boldsymbol{q}_{(n-1)i}[k]$. When $m$ values shared by selfish clients are larger than $\boldsymbol{q}_{0i}[k]$, the median reaches its maximum; when $m$ values are smaller than $\boldsymbol{q}_{(n-1)i}[k]$, the median is minimized. Hence, the upper and lower bounds are:

$$\overline{\boldsymbol{w}}_i[k] = \frac{1}{2}(\boldsymbol{q}_{\lfloor\frac{n-m-1}{2}\rfloor i}[k] + \boldsymbol{q}_{\lfloor\frac{n-m}{2}\rfloor i}[k]), \quad \underline{\boldsymbol{w}}_i[k] = \frac{1}{2}(\boldsymbol{q}_{\lfloor\frac{n+m-1}{2}\rfloor i}[k] + \boldsymbol{q}_{\lfloor\frac{n+m}{2}\rfloor i}[k]). \tag{5}$$

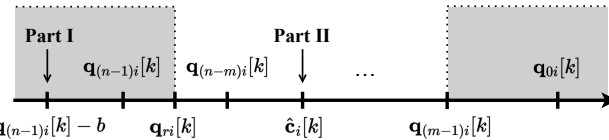

Figure 2: Example of selfish clients crafting shared models to attack Trimmed-mean. Values increase from left to right. Arrows indicate the shared model parameters of selfish clients. Parameters within the two grey regions will be filtered out by a non-selfish client.

**Shared model construction:** Our goal is to craft $m$ selfish clients' shared values such that the median equals $w_i^*[k]$. A simple strategy is to set all $m$ values to $w_i^*[k]$, which works when the median falls exactly at this value. However, when the total number of clients is even and $w_i^*[k]$ lies between the two middle elements of the aggregated sequence, the resulting median deviates. Specifically, if $w_i^*[k] \in [q_{ui}[k], \overline{w}_i[k]]$ or $[\underline{w}_i[k], q_{vi}[k]]$, where $u = \lfloor \frac{n-m}{2} \rfloor$ and $v = \lfloor \frac{n+m-1}{2} \rfloor$, the final median becomes the average of $w_i^*[k]$ and either $q_{ui}[k]$ or $q_{vi}[k]$, which is not the exactly $w_i^*[k]$. To address this, we set $m-1$ selfish clients values to $w_i^*[k]$ and adjust one value (e.g., $w_{(n,i)}[k]$), like the following:

$$w_{(n+1,i)}[k] = \cdots = w_{(n+m-1,i)}[k] = w_i^*[k],$$

$$w_{(n,i)}[k] = \begin{cases} 2w_i^*[k] - q_{ui}[k], & \text{if } w_i^*[k] > q_{ui}[k], \\ 2w_i^*[k] - q_{vi}[k], & \text{if } w_i^*[k] < q_{vi}[k], \\ w_i^*[k], & \text{otherwise.} \end{cases} \tag{6}$$

We prove that the above design guarantees that the overall median equals $w_i^*[k]$ in Appendix D.

### 4.6 Attacking Trimmed-mean

**Bound derivation:** In Trimmed-mean, each client drops the largest and smallest $c$ values per dimension and averages the remaining ones. We assume $c = m$, the number of selfish clients, which gives advantages to non-selfish clients. Given $m < N/3$, $w_i[k]$ has bounded range. Similar to attacking Median, denote $q_{0i}[k] \geq \cdots \geq q_{(n-1)i}[k]$ be the sorted values from non-selfish clients. If all $m$ selfish values are set above $q_{0i}[k]$ (or below $q_{(n-1)i}[k]$), they get trimmed, leading to the following bounds:

$$\overline{w}_i[k] = \frac{1}{n-m} \sum_{h=0}^{n-m-1} q_{hi}[k], \quad \underline{w}_i[k] = \frac{1}{n-m} \sum_{h=m}^{n-1} q_{hi}[k]. \tag{7}$$

**Shared model construction:** After computing $w_i^*[k]$, we design $m$ selfish models so that the trimmed mean equals $w_i^*[k]$. Our intuition is, to manipulate Trimmed-mean, selfish clients can divide their shared models into two groups: one deliberately crafted to be filtered out, and the other carefully adjusted to influence the final aggregation. By balancing the number and values of unfiltered shared models, the attacker ensures that the trimmed mean equals the desired target $w_i^*[k]$, while maintaining stealth under the aggregation rule's robustness. Therefore, we split the parameters of the $k$th dimension of selfish clients's shared models into two parts: *Part I selfish model parameters* are extreme values that get trimmed, and *Part II selfish model parameters* are values included in the aggregation, tuned to steer the average. We discuss two cases to craft the selfish models.

**Case I:** $w_i^*[k] \leq \tilde{w}_i[k]$. We set Part I selfish model parameters to be smaller than $q_{(n-1)i}[k]$ so they will be trimmed, and set remaining $(n-r)$ Part II values to:

$$c_i[k] = \frac{(n-m)w_i^*[k] - \sum_{h=m}^{r-1} q_{hi}[k]}{n-r}, \tag{8}$$

with $r$ is the largest integer to ensure $n - m \leq r \leq n$ and $w_i^*[k] \leq \frac{1}{n-m}((n-r) \cdot q_{(m-1)i}[k] + \sum_{h=m}^{r-1} q_{hi}[k])$. In summary, we have:

$$w_{(j,i)}[k] = \begin{cases} q_{(n-1)i}[k] - b, & 2n - r \leq j \leq n + m - 1 \\ c_i[k], & n \leq j \leq 2n - r - 1 \end{cases}, \tag{9}$$

where $b$ is a positive constant to guarantee that $w_{(j,i)}[k]$ is smaller than $q_{(n-1)i}[k]$, e.g., $b = 1$ in our experiments. Figure 2 shows an example of crafting selfish clients' shared models.

**Case II:** $\boldsymbol{w}_i^*[k] > \tilde{\boldsymbol{w}}_i[k]$. In this case, we aim to increase the trimmed mean. We set Part I selfish model parameters to be larger than $\boldsymbol{q}_{0i}[k]$ so they will be trimmed, and set the remaining $(r+1)$ Part II values to:

$$\boldsymbol{c}_i[k] = \frac{(n-m) \cdot \boldsymbol{w}_i^*[k] - \sum\limits_{h=r+1}^{n-m-1} \boldsymbol{q}_{hi}[k]}{r+1}. \tag{10}$$

where $r$ is the smallest integer such that $-1 \leq r \leq m-1$ and $\boldsymbol{w}_i^*[k] \geq \frac{1}{n-m}((r+1) \cdot \boldsymbol{q}_{(n-m)i}[k] + \sum_{h=r+1}^{n-m-1} \boldsymbol{q}_{hi}[k])$. This ensures that Part II values are not filtered out. Formally, we construct:

$$\boldsymbol{w}_{(j,i)}[k] = \begin{cases} \boldsymbol{q}_{0i}[k] + b, & n+r+1 \leq j \leq n+m-1 \\ \boldsymbol{c}_i[k], & n \leq j \leq n+r \end{cases}. \tag{11}$$

where $b$ is a positive constant to guarantee that $\boldsymbol{w}_{(j,i)}[k]$ is larger than $\boldsymbol{q}_{0i}[k]$.

This design ensures all Part I selfish model parameters are trimmed and Part II parameters dominate the aggregation, yielding $\boldsymbol{w}_i[k] = \boldsymbol{w}_i^*[k]$. See Appendix E for detailed proof.

### 4.7 Attacking Other Aggregation Rules

For non-coordinate-wise aggregation rules (e.g., Krum, FLTrust, FLDetector, RFA, and FLAME), exact optimal shared models are difficult to derive. These rules treat model parameters holistically using non-differentiable mechanisms like Euclidean distance scoring (e.g., FLTrust, FLDetector, RFA) or clustering-based aggregation (e.g., Krum, FLAME), making it hard to optimize per-dimension as we do for coordinate-wise rules. To demonstrate the transferability of SelfishAttack and its robustness against unknown rules, selfish clients can still use shared models crafted for FedAvg, Median, or Trimmed-mean (we use FedAvg in experiments). We also tailor our attack for FLAME, as described in Appendix I. While these attacks may not be optimal, experiments show they remain effective.

### 4.8 When to Start Attack

If selfish clients attack too early, non-selfish clients may fail to learn good local models, limiting the usefulness of the shared models and hurting the utility goal. To avoid this, we design a mechanism to determine when selfish clients should start attacking. The core idea is to wait until clients have learned sufficiently accurate local models. In each global round $t$, selfish clients share their local training losses $l_j^{(t)}$. Each client then computes the average loss and tracks the minimum value $l^{(t)}$ over time. When the decrease in $l^{(t)}$ over the past $I$ rounds becomes small—specifically, less than an $\epsilon$ fraction of the maximum decrease observed—selfish clients begin attacking by sending crafted models. This indicates that training has stabilized and clients likely have decent local models. The timing is controlled by two hyperparameters, $\epsilon$ and $I$, whose effects are analyzed in Section J.

## 5 Evaluation

### 5.1 Experimental Setup

**Datasets:** We evaluate the proposed SelfishAttack on three benchmark datasets—CIFAR-10 [20] and FEMNIST [4] for image classification, and Sent140 [16] for text classification. Detailed dataset descriptions are provided in Appendix F.

**Aggregation rules:** A client can use different aggregation rules to aggregate the shared models as a post-aggregation local model. We evaluate SelfishAttack under a diverse set of aggregation rules, including both standard and Byzantine-robust methods: FedAvg [24], Median [36], Trimmed-mean [36], Krum [3], FLTrust [5], FLDetector [37], RFA [27], and FLAME [25]. Detailed descriptions of these rules are provided in Appendix G.

**DFL parameter settings:** We assume 20 clients by default, including 6 selfish and 14 non-selfish ones. Clients train a CNN on CIFAR-10 and FEMNIST, and an LSTM [18] on Sent140. The CNN and LSTM architectures are shown in Tables 5a, 5b, and 6 in Appendix. Training hyperparameters are summarized in Table 7 in Appendix. We simulate non-IID for CIFAR-10 following [10] with

Table 1: Results (MTAS/MTANS/Gap) under different aggregation rules across datasets.

| Dataset | Method | FedAvg | Median | Trimmed-mean |
|---|---|---|---|---|
| CIFAR-10 | No attack | 0.627 / 0.627 / 0.000 | 0.644 / 0.644 / 0.000 | 0.644 / 0.644 / 0.000 |
| | Independent | 0.321 / 0.307 / 0.014 | 0.321 / 0.307 / 0.014 | 0.321 / 0.307 / 0.014 |
| | Two Coalitions | 0.505 / 0.578 / -0.073 | 0.505 / 0.578 / -0.073 | 0.505 / 0.578 / -0.073 |
| | TrimAttack | 0.207 / 0.165 / 0.043 | 0.524 / 0.513 / 0.011 | 0.456 / 0.440 / 0.016 |
| | GaussianAttack | 0.098 / 0.089 / 0.008 | 0.619 / 0.617 / 0.001 | 0.613 / 0.609 / 0.004 |
| | SelfishAttack | 0.475 / 0.326 / **0.150** | 0.543 / 0.425 / **0.119** | 0.597 / 0.484 / **0.113** |
| FEMNIST | No attack | 0.790 / 0.790 / 0.000 | 0.791 / 0.791 / 0.000 | 0.795 / 0.795 / 0.000 |
| | Independent | 0.543 / 0.545 / -0.002 | 0.543 / 0.545 / -0.002 | 0.543 / 0.545 / -0.002 |
| | Two Coalitions | 0.755 / 0.784 / -0.029 | 0.755 / 0.784 / -0.029 | 0.755 / 0.784 / -0.029 |
| | TrimAttack | 0.056 / 0.055 / 0.001 | 0.751 / 0.741 / 0.010 | 0.744 / 0.736 / 0.008 |
| | GaussianAttack | 0.003 / 0.028 / -0.025 | 0.788 / 0.785 / 0.003 | 0.793 / 0.789 / 0.003 |
| | SelfishAttack | 0.697 / 0.575 / **0.123** | 0.771 / 0.643 / **0.128** | 0.792 / 0.747 / **0.045** |
| Sent140 | No attack | 0.791 / 0.791 / 0.000 | 0.824 / 0.824 / 0.000 | 0.788 / 0.788 / 0.000 |
| | Independent | 0.631 / 0.637 / -0.006 | 0.631 / 0.637 / -0.006 | 0.631 / 0.637 / -0.006 |
| | Two Coalitions | 0.751 / 0.791 / -0.039 | 0.751 / 0.791 / -0.039 | 0.751 / 0.791 / -0.039 |
| | TrimAttack | 0.740 / 0.715 / 0.025 | 0.760 / 0.763 / -0.003 | 0.771 / 0.765 / 0.006 |
| | GaussianAttack | 0.536 / 0.528 / 0.008 | 0.807 / 0.807 / 0.000 | 0.785 / 0.785 / 0.000 |
| | SelfishAttack | 0.799 / 0.647 / **0.152** | 0.804 / 0.683 / **0.121** | 0.816 / 0.730 / **0.085** |

$\rho = 0.7$ (details in Appendix H). FEMNIST and Sent140 are naturally non-IID, so no simulation is needed. All the experiments are finished on one single Quadro RTX 6000 GPU with 24GB memory.

**Parameter settings for SelfishAttack:** By default, we set $\lambda = 0.0$ for FedAvg, $\lambda = 0.5$ for Median, and $\lambda = 1.0$, $b = 1.0$ for Trimmed-mean, as they use different aggregation strategies. Since selfish clients know the rule used in DFL, they can adjust $\lambda$ accordingly. We fix $\epsilon = 0.1$ and $I = 50$ across all aggregation rules, and report CIFAR-10 results unless stated otherwise. By default, selfish clients adopt the same aggregation rule and use all shared models, but we also explore cases where they use different aggregation rules or only their own shared models for aggregation.

**Evaluation metrics:** We use three metrics: *mean test accuracy of selfish clients (MTAS)*, *mean test accuracy of non-selfish clients (MTANS)*, and their difference (*Gap = MTAS - MTANS*). MTAS and MTANS are the average accuracies of selfish and non-selfish clients' local models:

$$\text{MTAS} = \frac{1}{m} \sum_{j=n}^{n+m-1} \text{ACC}(\boldsymbol{w}j, \mathcal{D}), \quad \text{MTANS} = \frac{1}{n} \sum i = 0^{n-1} \text{ACC}(\boldsymbol{w}_i, \mathcal{D}),$$

where $\text{ACC}(\boldsymbol{w}_i, \mathcal{D})$ denotes client $i$'s test accuracy with local model $\boldsymbol{w}_i$ on testing set $\mathcal{D}$. Note that selfish clients share the same final model, while non-selfish ones may differ.

## 5.2 Compared Methods

**Independent.** Each client independently trains a local model using its local data without DFL.

**Two Coalitions.** This method divides the clients into two coalitions: selfish clients and non-selfish clients. Clients within the same coalition collaborate using DFL with the FedAvg aggregation rule. This scenario means that selfish clients do not join DFL together with non-selfish clients.

**TrimAttack [10].** TrimAttack was originally designed as a model poisoning attack to CFL. We extend it to our problem setting. Specifically, selfish clients use TrimAttack to craft shared models sent to non-selfish clients such that the post-aggregation local model of a non-selfish client after attack deviates substantially from the one before attack, in terms of both direction and magnitude.

**GaussianAttack [3].** A selfish client sends a random Gaussian vector as a shared model to a non-selfish client. Each dimension of the Gaussian vector is generated from a normal distribution with zero mean and standard deviation 200.

## 5.3 Main Results

**SelfishAttack achieves both attack goals:** Table 1 summarizes the performance of SelfishAttack and baselines across different datasets and aggregation rules. "No attack" indicates selfish clients behave honestly by sending their own local models. In summary, our results demonstrate that SelfishAttack successfully achieves both the utility and competitive-advantage goals.

First, selfish clients achieve higher accuracy when collaborating with non-selfish clients under SelfishAttack, compared to training only among themselves (Two Coalitions), confirming the utility goal. For example, under Median and Trimmed-mean, MTASs of SelfishAttack are consistently higher than those of Two Coalitions across all datasets. The exception is FedAvg on CIFAR-10/FEMNIST. This is because FedAvg is not robust, in which the post-aggregation local models of non-selfish clients are negatively influenced by shared models from selfish clients too much.

Second, SelfishAttack effectively increases the gap between the mean accuracy of selfish and non-selfish clients, achieving the competitive-advantage goal. Particularly, the Gap exceeds $10\%$ in most cases, showing a clear competitive advantage. On CIFAR-10, for instance, Gaps reach 15.0%, 11.9%, and 11.3% under FedAvg, Median, and Trimmed-mean, respectively.

**SelfishAttack consistently outperforms existing attacks:** TrimAttack and GaussianAttack produce negligible or even negative Gaps in most cases, especially under robust rules like Median. GaussianAttack performs poorly under robust aggregation rules like Median and Trimmed-mean, and achieves low MTAS under FedAvg. In contrast, SelfishAttack allows selfish clients to benefit from useful shared models provided by non-selfish clients, improving their local models. TrimAttack fails in this regard, as non-selfish clients learn inaccurate models and propagate poor updates, resulting in low accuracy for both selfish and non-selfish clients, and thus smaller Gaps.

**Aggregating shared models from all vs. selfish clients only:** By default, selfish clients aggregate shared models from both selfish and non-selfish clients. However, attacks like TrimAttack and GaussianAttack can degrade their local model accuracy,

Table 2: MTAS/MTANS/Gap when a selfish client aggregates all vs. only selfish clients' models.

| Info. | TrimAttack | GaussianAttack | SelfishAttack |
|---|---|---|---|
| All | 0.456 / 0.440 / 0.016 | 0.613 / 0.609 / **0.004** | 0.597 / 0.484 / **0.113** |
| Selfish | 0.540 / 0.440 / **0.100** | 0.540 / 0.609 / -0.069 | 0.540 / 0.484 / 0.057 |

as they hinder non-selfish clients' learning and introduce harmful shared models into aggregation. To avoid this, selfish clients may choose to aggregate only models from selfish clients. In Table 2, "All" refers to aggregating from all clients, and "Selfish" means using only selfish clients' models. Selfish clients use FedAvg to enhance accuracy, while non-selfish clients use Trimmed-mean for robustness. SelfishAttack benefits from aggregating all shared models, as it still preserves useful information from non-selfish clients. In contrast, TrimAttack performs better when selfish clients exclude non-selfish models, which are degraded under attack. Overall, the best strategy is to use SelfishAttack and aggregate from all clients.

## 5.4 Ablation Studies

This section presents ablation studies on SelfishAttack, examining the impact of $\lambda$, degree of non-IID data, fraction of selfish clients, and other aggregation rules. Appendix J further explores the impact of $\epsilon$ and $I$, total number of clients, and aggregation rule used by selfish clients.

**Impact of $\lambda$:** Fig. 3 shows how $\lambda$ in Equation 1 affects the Gap under different aggregation rules. For robust aggregation rules like Median and Trimmed-mean, increasing $\lambda$ initially increases the Gap, as more emphasis is placed on the competitive-advantage goal. For example, with Trimmed-mean on CIFAR-10, the Gap rises from 6.2% at $\lambda$=0.5 to 11.3% at $\lambda$=1.0. However, beyond a threshold, e.g., $\lambda$>0.5 for Median or 1.0 for Trimmed-mean, the Gap drops because overly degraded non-selfish models harm the selfish clients' local models in later rounds.

**Impact of Non-IID degree:** Fig. 4 shows that SelfishAttack consistently outperforms other attacks on CIFAR-10 across various Non-IID levels and aggregation rules. For instance, when attacking Median, SelfishAttack maintains at least a 5% higher Gap than all baselines. Other attacks typically yield Gaps near zero, except for TrimAttack under FedAvg with moderate Non-IID (e.g., 0.5 or 0.7). As the Non-IID level increases, SelfishAttack becomes more effective, except for FedAvg where the Gap slightly declines from Non-IID 0.7 to 0.9 due to aggregation instability. Still, SelfishAttack achieves a 7.2% Gap under FedAvg at the highest Non-IID level.

**Impact of selfish client fraction:** Fig. 5 shows the impact of the selfish client fraction. SelfishAttack consistently outperforms other attacks across aggregation rules and different fractions. Compared attacks generally yield Gaps close to 0, except TrimAttack on FedAvg. For SelfishAttack, the Gap remains stable under FedAvg but increases with more selfish clients under Median

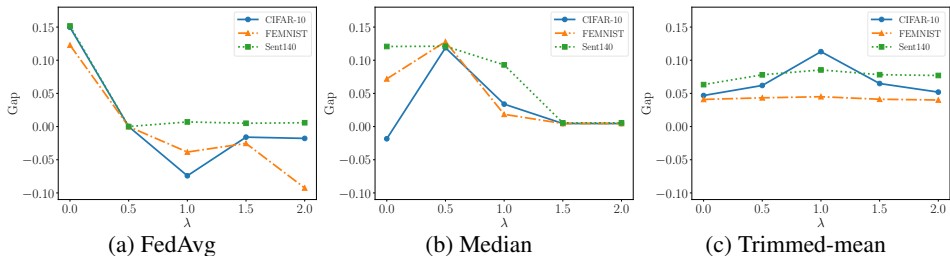

(a) FedAvg        (b) Median        (c) Trimmed-mean

Figure 3: Impact of $\lambda$ on Gap of SelfishAttack when DFL uses different aggregation rules.

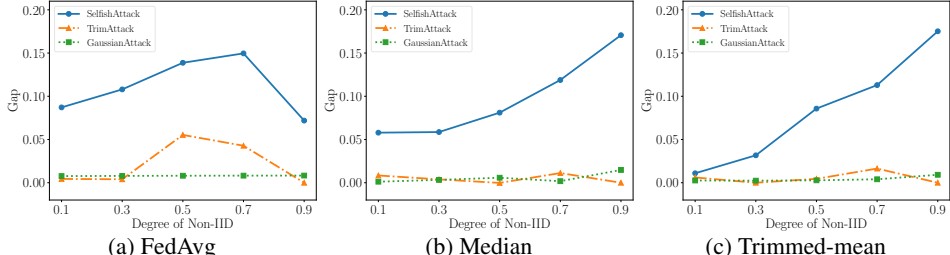

(a) FedAvg        (b) Median        (c) Trimmed-mean

Figure 4: Impact of the degree of Non-IID on Gap when DFL uses different aggregation rules.

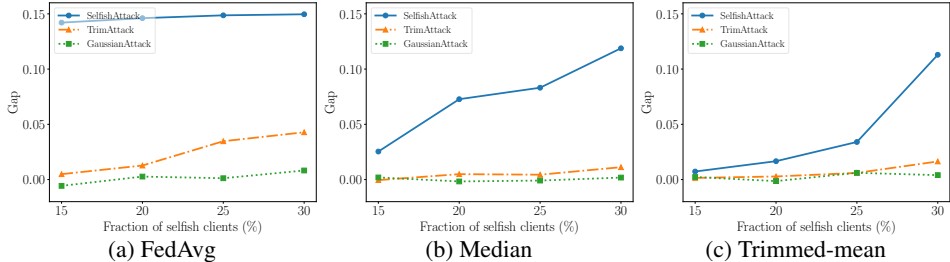

(a) FedAvg        (b) Median        (c) Trimmed-mean

Figure 5: Impact of the fraction of selfish clients on Gap when DFL uses different aggregation rules.

and Trimmed-mean. This is because more selfish clients enlarge the upper and lower bounds in Equations 5 and 7, expanding the attack range and improving effectiveness.

**Attacking other aggregation rules:** Table 3 shows that SelfishAttack remains effective when non-selfish clients use Krum, FLTrust, FLDetector, RFA, or FLAME. Selfish clients apply a tailored version for FLAME (detailed in Appendix I), or the FedAvg-based version of SelfishAttack for other aggregation rules. In all cases, SelfishAttack achieves notable Gaps (e.g., 18.9% for Krum, 15.1% for RFA, and 10.5% for FLAME), indicating SelfishAttack can transfer to these aggregation rules.

Table 3: Results of attacking other aggregation rules used by the non-selfish clients.

| Aggregation Rule | MTAS | MTANS | Gap |
|---|---|---|---|
| Krum | 0.545 | 0.356 | 0.189 |
| FLTrust | 0.537 | 0.469 | 0.069 |
| FLDetector | 0.556 | 0.439 | 0.118 |
| RFA | 0.484 | 0.333 | 0.151 |
| FLAME | 0.525 | 0.420 | 0.105 |

## 6 Conclusion, Limitations, and Future Work

In this paper, we propose SelfishAttack, the first competitive advantage attack for DFL. In SelfishAttack, selfish clients craft shared models to 1). learn more accurate local models than performing DFL among themselves, and 2). outperform non-selfish clients. These models are generated by solving an optimization problem that balances two attack goals. Experiments on three benchmarks show that SelfishAttack achieves both attack goals and outperforms conventional poisoning attacks. However, while SelfishAttack is effective across various aggregation rules, its optimality is unknown for non-coordinate-wise rules (e.g., Krum, FLAME). Future work includes developing attacks that adapt to unknown or dynamic aggregation rules, extending SelfishAttack to broader settings, and designing defenses against SelfishAttack.

## Acknowledgement

We thank the anonymous reviewers for their constructive comments. This work was supported by NSF under grant no. 2131859, 2125977, 2112562, and 1937787.

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

# Appendix

## A  Preliminaries and Related Work

### A.1  Decentralized Federated Learning

Consider a decentralized federated learning (DFL) system with $N$ clients. Each client $h$ has a local training dataset $D_h$, where $h = 0, 1, \cdots, N - 1$. These $N$ clients collaboratively train a machine learning model without the reliance of a central server. In particular, DFL aims to find a model $\boldsymbol{w}$ that minimizes the weighted average of losses among all clients: $\min_{\boldsymbol{w} \in \mathbb{R}^d} \frac{1}{N} \sum_{h=0}^{N-1} F_h(\boldsymbol{w}, \mathcal{D}_h)$, where $F_h(\boldsymbol{w}, \mathcal{D}_h) = \frac{1}{|\mathcal{D}_h|} \sum_{\zeta \in \mathcal{D}_h} F(\boldsymbol{w}, \zeta)$ is the local training objective of client $h$, $d$ is the number of model parameters, and $|\mathcal{D}_h|$ is the number of training examples of client $h$. In each *global training round*, DFL performs the following three steps:

- **Step I.** Every client trains a local model using its own local training dataset. Specifically, for client $h$, it samples a mini-batch of training examples from its local training dataset, and calculates a stochastic gradient $\boldsymbol{g}_h$. Then, client $h$ updates its local model as $\hat{\boldsymbol{w}}_h \leftarrow \breve{\boldsymbol{w}}_h - \eta \cdot \boldsymbol{g}_h$, where $\eta$ represents the learning rate, $\breve{\boldsymbol{w}}_h$ is the local model of client $h$ at the beginning of the current global training round, and $\hat{\boldsymbol{w}}_h$ is the *pre-aggregation local model* of client $h$ after local training. Note that client $h$ can compute stochastic gradient and update its local model multiple times in each global training round.

- **Step II.** Client $h$ sends a *shared model* $\boldsymbol{w}_{(h,h')}$ to each client $h'$, where $0 \leq h' \leq N - 1$. For notation convenience, we assume a client $h$ sends its own pre-aggregation local model to itself, i.e., $\boldsymbol{w}_{(h,h)} = \hat{\boldsymbol{w}}_h$. In non-adversarial settings, the shared model $\boldsymbol{w}_{(h,h')}$ is the pre-aggregation local model of client $h$, i.e., $\boldsymbol{w}_{(h,h')} = \hat{\boldsymbol{w}}_h$. SelfishAttack carefully crafts the shared models sent from selfish clients to non-selfish ones.

- **Step III.** Client $h$ aggregates the clients' shared models and updates its local model as $\boldsymbol{w}_h = \mathrm{Agg}(\{\boldsymbol{w}_{(h',h)}\}_{0 \leq h' \leq N-1})$, where $\{\boldsymbol{w}_{(h',h)}\}_{0 \leq h' \leq N-1}$ is the set of shared models other clients sent to $h$ and $\mathrm{Agg}(\cdot)$ denotes an aggregation rule. We call $\boldsymbol{w}_h$ the *post-aggregation local model* of client $h$ at the end of the current global training round.

DFL repeats the above iterative process for multiple global training rounds. Different DFL methods use different aggregation rules, such as FedAvg [23] and Median [36]. Note that in non-adversarial settings, all clients have the same post-aggregation local model in each global training round. Table 4 summarizes the key notations used in our paper.

### A.2  Poisoning Attacks to CFL/DFL

Conventional poisoning attacks to FL can be divided into two categories depending on the goal of the attacker, namely *untargeted attacks* [3, 10] and *targeted attacks* [1, 2]. These poisoning attacks were originally designed for CFL, but they can be extended to our problem setting. Specifically, an untargeted attack aims to manipulate the DFL system such that a poisoned local model of a non-selfish client produces incorrect predictions for a large portion of clean testing inputs indiscriminately, i.e., the final learnt local model of a non-selfish client has a low testing accuracy. For instance, in Gaussian attack [3], each selfish client sends an arbitrary Gaussian vector to non-selfish clients; while Trim attack [10] carefully crafts the shared models sent from selfish clients to non-selfish ones in order to significantly deviate non-selfish clients' post-aggregation local models in each global training round. A targeted attack [1, 2] aims to poison the system such that the local model of a non-selfish client predicts a predefined label for inputs that contain special characteristics such as those embedded with predefined triggers.

### A.3  Byzantine-robust Aggregation Rules

The FedAvg [23] aggregation rule is commonly used in non-adversarial settings. However, a single shared model can arbitrarily manipulate the post-aggregation local model of a non-selfish client in FedAvg [3]. Byzantine-robust aggregation rules [3, 5, 25, 36] aim to be robust against "outlier" shared models. In particular, when a non-selfish client uses a Byzantine-robust aggregation rule,

**Algorithm 1** SelfishAttack Algorithm. Lines 5 to 7 describe Step I of DFL, where each client performs local training of its pre-aggregation local model. Lines 8 to 13 illustrate the process of exchanging shared models among clients in the global training rounds before our attack starts. Lines 16 to 22 provide a detailed description of Section 4.8, explaining how to determine when to start attack. The process of performing attack is shown in lines 24 and 25, while the methods of crafting shared models are presented in Algorithm 2.

---

**Input:** Number of non-selfish clients $n$; number of selfish clients $m$; loss checking interval $I$; parameter $\lambda$ in loss function; parameter $\epsilon$; aggregation rule $\text{Agg}(\cdot)$; and total number of global training rounds $T$.

1: $start$=False;
2: $l^{(0)} = Inf$;
3: $max\_gap = 0$;
4: **for** $t = 1$ to $T$ **do**
5:   **for** $h = 0$ to $n + m - 1$ **do**             ▷ all clients
6:    $\hat{\boldsymbol{w}}_h, l_h^{(t)} = LocalUpdate(\check{\boldsymbol{w}}_h, D_h)$;
7:   **end for**
8:   **for** $i = 0$ to $n - 1$ **do**           ▷ all non-selfish clients
9:    Client $i$ sends $\boldsymbol{w}_{(i,h)} = \hat{\boldsymbol{w}}_i$ to each client $h$;
10:   **end for**
11:   **for** $j = n$ to $n + m - 1$ **do**          ▷ all selfish clients
12:    Client $j$ sends $\boldsymbol{w}_{(j,h)} = \hat{\boldsymbol{w}}_j$ and $l_j^{(t)}$ to each selfish client $h$;
13:   **end for**
14:   **for** $j = n$ to $n + m - 1$ **do**          ▷ all selfish clients
15:    Client $j$ receives $\boldsymbol{W}_j = \{\boldsymbol{w}_{(i,j)}\}_{0 \le i \le n-1}$;
16:    $l^{(t)} = \min(l^{(t-1)}, \frac{1}{m}\sum_{n \le j \le n+m-1} l_j^{(t)})$;
17:    **if** $t > I$ and $start$==False **then**
18:     $max\_gap = \max(l^{(t-I)} - l^{(t)}, max\_gap)$;
19:     **if** $0 < l^{(t-I)} - l^{(t)} < \epsilon \cdot max\_gap$ **then**
20:      $start$=True;
21:     **end if**
22:    **end if**
23:    **for** $i = 0$ to $n - 1$ **do**
24:     $\boldsymbol{w}_{(j,i)} = Sel(n, m, \boldsymbol{W}_j, \hat{\boldsymbol{w}}_j, \text{Agg}(\cdot), i, \lambda, start)$;
25:     Client $j$ sends $\boldsymbol{w}_{(j,i)}$ to non-selfish client $i$;
26:    **end for**
27:   **end for**
28:   **for** $h = 0$ to $n + m - 1$ **do**          ▷ all clients
29:    $\boldsymbol{w}_h = \text{Agg}(\{\boldsymbol{w}_{(h',h)}\}_{0 \le h' \le n+m-1})$;
30:    $\check{\boldsymbol{w}}_h = \boldsymbol{w}_h$;
31:   **end for**
32: **end for**

---

its post-aggregation local model is less likely to be influenced by outlier shared models, e.g., those from selfish clients.

For instance, Median [36] and Trimmed-mean [36] are two coordinate-wise aggregation rules, which remove the outliers in each dimension of clients' shared models in order to reduce the impact of outlier shared models. Specifically, for a given client, the Median aggregation rule outputs the coordinate-wise median of the client's received shared models as the post-aggregation local model. For each dimension, Trimmed-mean first removes the largest $c$ and smallest $c$ elements, then takes the average of the remaining $N - 2c$ items, where $c$ is the trim parameter. Krum [3] aggregates a client's received shared models by selecting the shared model that has the smallest sum of Euclidean distance to its subset of neighboring shared models. We note that the previous work [9] has already applied Median, Trimmed-mean, and Krum aggregation rules in the context of DFL. In FLTrust [5], if a received shared model diverges significantly from the pre-aggregation local model of client $i$, then client $i$ assigns a low trust score to this received shared model. In FLDetector [37], clients leverage clustering techniques to detect outlier shared models and then aggregate shared models from clients that have been detected as non-selfish. In RFA [27], clients use a geometric-median based robust aggregation oracle to aggregate shared models from other clients. In FLAME [25], clients leverage clustering, model weight clipping, and noise injection techniques to mitigate the impact of outlier shared models.

**Algorithm 2** $Sel(n, m, \boldsymbol{W}_j, \hat{\boldsymbol{w}}_j, \mathrm{Agg}(\cdot), i, \lambda, start)$.

---

**Input:** Number of non-selfish clients $n$; number of selfish clients $m$; non-selfish clients' local models $\boldsymbol{W}_j = \{\boldsymbol{w}_{(h,j)}\}_{0 \le h \le n-1}$; pre-aggregation local model $\hat{\boldsymbol{w}}_j$ of selfish client $j$; aggregation rule $\mathrm{Agg}(\cdot)$; non-selfish client index $i$; parameter $\lambda$ in loss function; and boolean variable $start$.
**Output:** Shared model $\boldsymbol{w}_{(j,i)}$ sent from client $j$ to $i$.
 1: **if** $start==\mathrm{False}$ **then**
 2:     $\boldsymbol{w}_{(j,i)} = \hat{\boldsymbol{w}}_j$;
 3:     **return** $\boldsymbol{w}_{(j,i)}$.
 4: **end if**
 5: **if** $\mathrm{Agg}(\cdot)$ is FedAvg **then**
 6:     **for** $k = 0$ to $d - 1$ **do**
 7:         Compute $\boldsymbol{w}_{(j,i)}[k]$ based on Equation 4;
 8:     **end for**
 9: **else if** $\mathrm{Agg}(\cdot)$ is Median **then**
10:     **for** $k = 0$ to $d - 1$ **do**
11:         Compute $\boldsymbol{w}_{(j,i)}[k]$ based on Equation 6;
12:     **end for**
13: **else if** $\mathrm{Agg}(\cdot)$ is Trimmed-mean **then**
14:     **for** $k = 0$ to $d - 1$ **do**
15:         Compute $\boldsymbol{w}_{(j,i)}[k]$ based on Equation 9 or 11;
16:     **end for**
17: **end if**
18: **return** $\boldsymbol{w}_{(j,i)}$.

---

## B  Solving the Quadratic Program (Equation 2)

**Theorem 1.** *The optimal solution $\boldsymbol{w}_i^*[k]$ of Equation 2 is as follows:*

$$\boldsymbol{w}_i^*[k] = \begin{cases} \boldsymbol{p}_i[k], & \lambda < 1 \wedge \boldsymbol{p}_i[k] \in [\underline{\boldsymbol{w}}_i[k], \overline{\boldsymbol{w}}_i[k]] \\ \underline{\boldsymbol{w}}_i[k] \text{ or } \overline{\boldsymbol{w}}_i[k], & otherwise \end{cases} . \tag{12}$$

*Proof.* We can solve $\boldsymbol{w}_i^*[k]$ separately in the following three cases:

*Case I:* $\lambda < 1$. In this case, if $\boldsymbol{p}_i[k]$ is within the interval $[\underline{\boldsymbol{w}}_i[k], \overline{\boldsymbol{w}}_i[k]]$, then the optimal solution is $\boldsymbol{w}_i^*[k] = \boldsymbol{p}_i[k]$. Otherwise, we have $\boldsymbol{w}_i^*[k] = \overline{\boldsymbol{w}}_i[k]$ if $\boldsymbol{p}_i[k]$ is larger than $\overline{\boldsymbol{w}}_i[k]$; and $\boldsymbol{w}_i^*[k] = \underline{\boldsymbol{w}}_i[k]$ if $\boldsymbol{p}_i[k]$ is smaller than $\underline{\boldsymbol{w}}_i[k]$. In other words, we have the following:

$$\boldsymbol{w}_i^*[k] = \begin{cases} \boldsymbol{p}_i[k], & \underline{\boldsymbol{w}}_i[k] \le \boldsymbol{p}_i[k] \le \overline{\boldsymbol{w}}_i[k] \\ \overline{\boldsymbol{w}}_i[k], & \boldsymbol{p}_i[k] > \overline{\boldsymbol{w}}_i[k] \\ \underline{\boldsymbol{w}}_i[k], & \boldsymbol{p}_i[k] < \underline{\boldsymbol{w}}_i[k] \end{cases} . \tag{13}$$

*Case II:* $\lambda = 1$. In this case, we have $\boldsymbol{w}_i^*[k]$ as follows:

$$\boldsymbol{w}_i^*[k] = \begin{cases} \overline{\boldsymbol{w}}_i[k], & \boldsymbol{w}_i[k] > \tilde{\boldsymbol{w}}_i[k] \\ \underline{\boldsymbol{w}}_i[k], & \boldsymbol{w}_i[k] \le \tilde{\boldsymbol{w}}_i[k] \end{cases} . \tag{14}$$

*Case III:* $\lambda > 1$. In this case, $\boldsymbol{w}_i^*[k]$ is a value between $\overline{\boldsymbol{w}}_i[k]$ and $\underline{\boldsymbol{w}}_i[k]$ that has the larger distance to $\boldsymbol{p}_i[k]$. That is, we have the following:

$$\boldsymbol{w}_i^*[k] = \begin{cases} \overline{\boldsymbol{w}}_i[k], & |\boldsymbol{p}_i[k] - \overline{\boldsymbol{w}}_i[k]| > |\boldsymbol{p}_i[k] - \underline{\boldsymbol{w}}_i[k]| \\ \underline{\boldsymbol{w}}_i[k], & |\boldsymbol{p}_i[k] - \overline{\boldsymbol{w}}_i[k]| \le |\boldsymbol{p}_i[k] - \underline{\boldsymbol{w}}_i[k]| \end{cases} . \tag{15}$$

By combining Equations 13-15, we can get Equation 12. □

## C  Proof of Attacking FedAvg

**Theorem 2.** *Suppose non-selfish client $i$ uses the FedAvg aggregation rule. The crafted shared models in Equation 4 for the selfish clients are optimal solutions to the optimization problem in Equation 1.*

*Proof.* According to Equation 4, we have:

$$\boldsymbol{w}_i[k] = \frac{1}{n+m} \sum_{h=0}^{n+m-1} \boldsymbol{w}_{(h,i)}[k]$$

$$= \frac{1}{n+m}(\sum_{h=0}^{n-1} \boldsymbol{w}_{(h,i)}[k] + (n+m)\boldsymbol{w}_i^*[k] - \sum_{h=0}^{n-1} \boldsymbol{w}_{(h,i)}[k]) \quad (16)$$

$$= \frac{1}{n+m}((n+m)\boldsymbol{w}_i^*[k])$$

$$= \boldsymbol{w}_i^*[k]$$

$\square$

## D   Proof of Attacking Median

**Theorem 3.** *Suppose non-selfish client $i$ uses the Median aggregation rule. The crafted shared models in Equation 6 for the selfish clients are optimal solutions to the optimization problem in Equation 1.*

*Proof.* We discuss the three cases in Equation 6 separately.

*Case I:* $\overline{\boldsymbol{w}}_i[k] \geq \boldsymbol{w}_i^*[k] > \boldsymbol{q}_{\lfloor \frac{n-m}{2} \rfloor i}[k]$. We have

$$\boldsymbol{q}_{\lfloor \frac{n-m-1}{2} \rfloor i}[k] \geq 2\boldsymbol{w}_i^*[k] - \boldsymbol{q}_{\lfloor \frac{n-m}{2} \rfloor i}[k] \geq \boldsymbol{q}_{\lfloor \frac{n-m}{2} \rfloor i}[k].$$

Thus

$$\boldsymbol{q}_{\lfloor \frac{n-m-1}{2} \rfloor i}[k] \geq \boldsymbol{w}_{(n,i)}[k] \geq \boldsymbol{q}_{\lfloor \frac{n-m}{2} \rfloor i}[k],$$

$$\boldsymbol{w}_{(n+1,i)}[k] = \cdots = \boldsymbol{w}_{(n+m-1,i)}[k] = \boldsymbol{w}_i^*[k] > \boldsymbol{q}_{\lfloor \frac{n-m-1}{2} \rfloor i}[k].$$

So $\boldsymbol{q}_{\lfloor \frac{n-m}{2} \rfloor i}[k]$ ranks $\lfloor \frac{n+m-1}{2} \rfloor$th among $\boldsymbol{w}_{(0,i)}[k], \cdots, \boldsymbol{w}_{(n+m-1,i)}[k]$, and $\boldsymbol{q}_{\lfloor \frac{n-m-1}{2} \rfloor i}[k], \boldsymbol{w}_{(n,i)}[k]$ rank $\lfloor \frac{n+m-1}{2} \rfloor$th, $\lfloor \frac{n+m+1}{2} \rfloor$th, respectively. We have

$$\boldsymbol{w}_i[k] = \frac{1}{2}(\boldsymbol{w}_{(n,i)}[k] + \boldsymbol{q}_{\lfloor \frac{n-m}{2} \rfloor i}[k]) = \boldsymbol{w}_i^*[k]. \quad (17)$$

*Case II:* $\boldsymbol{q}_{\lfloor \frac{n+m-1}{2} \rfloor i}[k] > \boldsymbol{w}_i^*[k] \geq \underline{\boldsymbol{w}}_i[k]$. We have

$$\boldsymbol{q}_{\lfloor \frac{n+m-1}{2} \rfloor i}[k] \geq 2\boldsymbol{w}_i^*[k] - \boldsymbol{q}_{\lfloor \frac{n+m-1}{2} \rfloor i}[k] \geq \boldsymbol{q}_{\lfloor \frac{n+m}{2} \rfloor i}[k].$$

Thus

$$\boldsymbol{q}_{\lfloor \frac{n+m-1}{2} \rfloor i}[k] \geq \boldsymbol{w}_{(n,i)}[k] \geq \boldsymbol{q}_{\lfloor \frac{n+m}{2} \rfloor i}[k],$$

$$\boldsymbol{w}_{(n+1,i)}[k] = \cdots = \boldsymbol{w}_{(n+m-1,i)}[k] = \boldsymbol{w}_i^*[k] < \boldsymbol{q}_{\lfloor \frac{n+m}{2} \rfloor i}[k].$$

So $\boldsymbol{q}_{\lfloor \frac{n+m-1}{2} \rfloor i}[k]$ ranks $\lfloor \frac{n+m+1}{2} \rfloor$th among $\boldsymbol{w}_{(0,i)}[k], \cdots, \boldsymbol{w}_{(n+m-1,i)}[k]$, and $\boldsymbol{q}_{\lfloor \frac{n+m}{2} \rfloor i}[k], \boldsymbol{w}_{(n,i)}[k]$ rank $\lfloor \frac{n+m+4}{2} \rfloor$th and $\lfloor \frac{n+m+2}{2} \rfloor$th, respectively. Therefore,

$$\boldsymbol{w}_i[k] = \frac{1}{2}(\boldsymbol{w}_{(n,i)}[k] + \boldsymbol{q}_{\lfloor \frac{n+m-1}{2} \rfloor i}[k]) = \boldsymbol{w}_i^*[k]. \quad (18)$$

*Case III:* $\boldsymbol{q}_{\lfloor \frac{n-m}{2} \rfloor i}[k] \geq \boldsymbol{w}_i^*[k] \geq \boldsymbol{q}_{\lfloor \frac{n+m-1}{2} \rfloor i}[k]$. In this case, we can find $r$ such that $\lfloor \frac{n+m-1}{2} \rfloor > r \geq \lfloor \frac{n-m}{2} \rfloor$ and $\boldsymbol{q}_{ri}[k] \geq \boldsymbol{w}_i^*[k] \geq \boldsymbol{q}_{(r+1)i}[k]$. Thus

$$\boldsymbol{q}_{ri}[k] \geq \boldsymbol{w}_{(n,i)}[k] = \cdots = \boldsymbol{w}_{(n+m-1,i)}[k] \geq \boldsymbol{q}_{(r+1)i}[k],$$

$\boldsymbol{q}_{ri}[k], \boldsymbol{w}_{(n,i)}[k], \cdots, \boldsymbol{w}_{(n+m-1,i)}[k], \boldsymbol{q}_{(r+1)i}[k]$ rank $(r+1)$th, $(r+2)$th, $\cdots, (r+m+1)$th, $(r+m+2)$th among $\boldsymbol{w}_{(0,i)}[k], \cdots, \boldsymbol{w}_{(n+m-1,i)}[k]$, respectively. Because $r+1 < \lfloor \frac{n+m+1}{2} \rfloor$ and $r+m+2 >$

$\lfloor \frac{n+m+2}{2} \rfloor$, the median value of $\boldsymbol{w}_{(0,i)}[k], \cdots, \boldsymbol{w}_{(n+m-1,i)}[k]$ must between $\boldsymbol{q}_{(r+1)i}[k]$ and $\boldsymbol{q}_{ri}[k]$, which means $\boldsymbol{q}_{ri}[k] \geq \boldsymbol{w}_i[k] \geq \boldsymbol{q}_{(r+1)i}[k]$. So we have

$$\boldsymbol{w}_i[k] = \frac{1}{2}(\boldsymbol{w}_i^*[k] + \boldsymbol{w}_i^*[k]) = \boldsymbol{w}_i^*[k]. \tag{19}$$

After summarizing results of Equations 17, 18, and 19, we have

$$\boldsymbol{w}_i[k] = \boldsymbol{w}_i^*[k].$$

$\square$

# E  Proof of Attacking Trimmed-mean

**Theorem 4.** *Suppose non-selfish client $i$ uses the Trimmed-mean aggregation rule. The crafted shared models in Equation 9 or Equation 11 for the selfish clients are optimal solutions to the optimization problem in Equation 1.*

*Proof.* We discuss two cases based on the definitions of Equation 9 and Equation 11.

Note that $\tilde{\boldsymbol{w}}_i[k] = \frac{1}{n-2m} \sum_{h=m}^{n-m-1} \boldsymbol{q}_{hi}[k]$.

*Case I: $\underline{\boldsymbol{w}}_i[k] \leq \boldsymbol{w}_i^*[k] \leq \tilde{\boldsymbol{w}}_i[k]$.* According to the definition of $r$, we need to find maximum $r$, such that $r \leq n$, and

$$\boldsymbol{w}_i^*[k] \leq \frac{1}{n-m}\left((n-r) \cdot \boldsymbol{q}_{(m-1)i}[k] + \sum_{h=m}^{r-1} \boldsymbol{q}_{hi}[k]\right). \tag{20}$$

Since $\boldsymbol{q}_{(m-1)i}[k] \geq \tilde{\boldsymbol{w}}_i[k] \geq \boldsymbol{w}_i^*[k]$, when $r = n - m$, we have

$$\begin{aligned}
(n-m) \cdot \boldsymbol{w}_i^*[k] &\leq m \cdot \boldsymbol{q}_{(m-1)i}[k] + (n-2m) \cdot \tilde{\boldsymbol{w}}_i[k] \\
&= m \cdot \boldsymbol{q}_{(m-1)i}[k] + \sum_{h=m}^{n-m-1} \boldsymbol{q}_{hi}[k] \\
&= (n-r) \cdot \boldsymbol{q}_{(m-1)i}[k] + \sum_{h=m}^{r-1} \boldsymbol{q}_{hi}[k],
\end{aligned} \tag{21}$$

so such $r$ exists, and $r \geq n - m$. If $r = n$, there are no Part II selfish model parameters, and

$$\boldsymbol{w}_i^*[k] \leq \frac{1}{n-m} \sum_{h=m}^{n-1} \boldsymbol{q}_{hi}[k] = \underline{\boldsymbol{w}}_i[k].$$

Since $\boldsymbol{w}_i^*[k] \geq \underline{\boldsymbol{w}}_i[k]$, we have $\boldsymbol{w}_i^*[k] = \underline{\boldsymbol{w}}_i[k]$. Note that in this case all the shared model parameters of selfish clients are Part I model parameters, which are all less than $\boldsymbol{q}_{(n-1)i}[k]$, so

$$\boldsymbol{w}_i[k] = \underline{\boldsymbol{w}}_i[k] = \boldsymbol{w}_i^*[k]. \tag{22}$$

If $r \leq n - 1$, then we prove that $\boldsymbol{q}_{ri}[k] \leq \boldsymbol{w}_{(j,i)}[k] \leq \boldsymbol{q}_{(m-1)i}[k]$ when $n \leq j \leq 2n - r - 1$. In fact, the right side can be directly obtained by Equation 20, so we only need to prove the left side. Since $r$ is maximum, we have

$$\boldsymbol{w}_i^*[k] \geq \frac{1}{n-m}\left((n-r-1) \cdot \boldsymbol{q}_{(m-1)i}[k] + \sum_{h=m}^{r} \boldsymbol{q}_{hi}[k]\right). \tag{23}$$

So we have

$$\begin{aligned}
\boldsymbol{w}_i^*[k] &\geq \frac{1}{n-m}\left((n-r-1) \cdot \boldsymbol{q}_{(m-1)i}[k] + \sum_{h=m}^{r} \boldsymbol{q}_{hi}[k]\right) \\
&\geq \frac{1}{n-m}\left((n-r-1) \cdot \boldsymbol{q}_{ri}[k] + \sum_{h=m}^{r} \boldsymbol{q}_{hi}[k]\right) \\
&= \frac{1}{n-m}\left((n-r) \cdot \boldsymbol{q}_{ri}[k] + \sum_{h=m}^{r-1} \boldsymbol{q}_{hi}[k]\right).
\end{aligned} \tag{24}$$

Thus $\boldsymbol{q}_{ri}[k] = \boldsymbol{c}_i[k] < \boldsymbol{w}_{(j,i)}[k]$ for $n \leq j \leq 2n - r - 1$ according to Equation 8. Hence, when calculating the trimmed mean value of $\{\boldsymbol{w}_{(h,i)}[k]\}_{0 \leq h \leq n+m-1}$, $\boldsymbol{q}_{0i}[k], \cdots, \boldsymbol{q}_{(m-1)i}[k]$ will be filtered out since they are the largest $m$ values in the $k$th dimension, and $\boldsymbol{q}_{ri}[k], \cdots, \boldsymbol{q}_{(n-1)i}[k], \boldsymbol{w}_{(2n-r,i)}[k], \cdots, \boldsymbol{w}_{(n+m-1,i)}[k]$ will be filtered out since they are the smallest $m$ values in the $k$th dimension, and

$$
\begin{aligned}
\boldsymbol{w}_i[k] &= \frac{1}{n-m} \Big( \sum_{h=m}^{r-1} \boldsymbol{q}_{hi}[k] + \sum_{j=n}^{2n-r-1} \boldsymbol{w}_{(j,i)}[k] \Big) \\
&= \frac{1}{n-m} \Big( \sum_{h=m}^{r-1} \boldsymbol{q}_{hi}[k] + (n-m) \cdot \boldsymbol{w}_i^*[k] - \sum_{h=m}^{r-1} \boldsymbol{q}_{hi}[k] \Big) \\
&= \frac{1}{n-m} \big( (n-m) \cdot \boldsymbol{w}_i^*[k] \big) \\
&= \boldsymbol{w}_i^*[k].
\end{aligned}
\tag{25}
$$

*Case II:* $\tilde{\boldsymbol{w}}_i[k] < \boldsymbol{w}_i^*[k] \leq \overline{\boldsymbol{w}}_i[k]$. Similar to Case I, according to the definition of $r$, first we need to find minimum $r$ such that $r \geq 0$, and

$$
\boldsymbol{w}_i^*[k] \geq \frac{1}{n-m} \Big( (r+1) \cdot \boldsymbol{q}_{(n-m)i}[k] + \sum_{h=r+1}^{n-m-1} \boldsymbol{q}_{hi}[k] \Big).
\tag{26}
$$

Since $\boldsymbol{q}_{(n-m)i}[k] \leq \tilde{\boldsymbol{w}}_i[k] \leq \boldsymbol{w}_i^*[k]$, when $r = m - 1$, we have

$$
\begin{aligned}
(n-m) \cdot \boldsymbol{w}_i^*[k] &\geq m \cdot \boldsymbol{q}_{(n-m)i}[k] + (n - 2m) \cdot \tilde{\boldsymbol{w}}_i[k] \\
&= m \cdot \boldsymbol{q}_{(n-m)i}[k] + \sum_{h=m}^{n-m-1} \boldsymbol{q}_{hi}[k] \\
&= (r+1) \cdot \boldsymbol{q}_{(n-m)i}[k] + \sum_{h=r+1}^{n-m-1} \boldsymbol{q}_{hi}[k],
\end{aligned}
\tag{27}
$$

so such $r$ exists, and $r \leq m - 1$. If $r = -1$, there are no Part II selfish model parameters and we have

$$
\boldsymbol{w}_i[k] = \overline{\boldsymbol{w}}_i[k] = \boldsymbol{w}_i^*[k].
\tag{28}
$$

If $r \geq 0$, then we can similarly prove $\boldsymbol{q}_{(n-m)i}[k] \leq \boldsymbol{w}_{(j,i)}[k] \leq \boldsymbol{q}_{ri}[k]$ when $n \leq j \leq n + r$. Hence, when calculating the trimmed mean value of $\{\boldsymbol{w}_{(h,i)}[k]\}_{0 \leq h \leq n+m-1}$, $\boldsymbol{q}_{(n-m)i}[k], \cdots, \boldsymbol{q}_{(n-1)i}[k]$ will be filtered out since they are the smallest $m$ values in the $k$th dimension, and $\boldsymbol{q}_{0i}[k], \cdots, \boldsymbol{q}_{ri}[k], \boldsymbol{w}_{(n+r+1,i)}[k], \cdots, \boldsymbol{w}_{(n+m-1,i)}[k]$ will be filtered out since they are the largest $m$ values in the $k$th dimension, and

$$
\begin{aligned}
\boldsymbol{w}_i[k] &= \frac{1}{n-m} \Big( \sum_{h=r+1}^{n-m-1} \boldsymbol{q}_{hi}[k] + \sum_{j=n}^{n+r} \boldsymbol{w}_{(j,i)}[k] \Big) \\
&= \frac{1}{n-m} \Big( \sum_{h=r+1}^{n-m-1} \boldsymbol{q}_{hi}[k] + (n-m)\boldsymbol{w}_i^*[k] - \sum_{h=r+1}^{n-m-1} \boldsymbol{q}_{hi}[k] \Big) \\
&= \frac{1}{n-m} \big( (n-m) \cdot \boldsymbol{w}_i^*[k] \big) \\
&= \boldsymbol{w}_i^*[k].
\end{aligned}
\tag{29}
$$

After summarizing results of Equations 22, 25, 28, and 29, we have:

$$
\boldsymbol{w}_i[k] = \boldsymbol{w}_i^*[k].
$$

$\square$

Table 4: Notations.

| Notation | Description |
|---|---|
| $n$ | Number of non-selfish clients. |
| $m$ | Number of selfish clients. |
| $h$ | Index of any clients. |
| $i$ | Index of non-selfish clients. |
| $j$ | Index of selfish clients. |
| $\lambda$ | Hyperparameter in our optimization problem. |
| $\check{\boldsymbol{w}}_i$ | Local model of client $i$ at the start of a global round. |
| $\hat{\boldsymbol{w}}_i$ | Pre-aggregation local model of client $i$. |
| $\boldsymbol{w}_i$ | Post-aggregation local model of client $i$. |
| $\boldsymbol{w}_i[k]$ | $k$th dimension of $\boldsymbol{w}_i$. |
| $\boldsymbol{w}_i^*$ | Optimal solution of our optimization problem. |
| $\tilde{\boldsymbol{w}}_i$ | Model aggregated by non-selfish clients' shared models. |
| $\overline{\boldsymbol{w}}_i[k]$ | Upper bound of $\boldsymbol{w}_i[k]$. |
| $\underline{\boldsymbol{w}}_i[k]$ | Lower bound of $\boldsymbol{w}_i[k]$. |
| $\boldsymbol{w}_{(h,h')}$ | Shared model that client $h$ sends to client $h'$. |

Table 5: CNN architectures for CIFAR-10 and FEMNIST.

(a) CIFAR-10.

| Layer | Size |
|---|---|
| Input | $32 \times 32 \times 3$ |
| Conv + ReLU | $3 \times 3 \times 30$ |
| Max Pooling | $2 \times 2$ |
| Conv + ReLU | $3 \times 3 \times 50$ |
| Max Pooling | $2 \times 2$ |
| FC + ReLU | 100 |
| FC | 10 |

(b) FEMNIST.

| Layer | Size |
|---|---|
| Input | $28 \times 28 \times 1$ |
| Conv + ReLU | $7 \times 7 \times 32$ |
| Max Pooling | $2 \times 2$ |
| Conv + ReLU | $3 \times 3 \times 64$ |
| Max Pooling | $2 \times 2$ |
| FC | 62 |

# F Dataset Description

**CIFAR-10 [20].** CIFAR-10 is a 10-class color image classification dataset, with 50,000 training examples and 10,000 testing examples. *License: MIT License. Available at `https://www.cs.toronto.edu/~kriz/cifar.html`.*

**Federated Extended MNIST (FEMNIST) [4].** FEMNIST is a 62-class image classification dataset. It is constructed by partitioning data from Extended MNIST [22] based on the writer of the digit or character. There are 3,550 writers and 805,263 examples in total. To distribute training examples to clients, we select 10 writers for each client and combine their data as the local training data for a client. *License: Apache License 2.0 (via LEAF benchmark). Available at `https://leaf.cmu.edu`.*

**Sentiment140 (Sent140) [16].** Sent140 is a two-class text classification dataset for sentiment analysis. The data are tweets collected from 660,120 Twitter users and annotated based on the emoticons present in themselves. In our experiments, we choose 300 users with at least 10 tweets for each client, and the union of the training tweets of these 300 users is the client's local training data. *License: Other. No explicit license is provided. Available for academic use only. Hosted on Hugging Face at `https://huggingface.co/datasets/stanfordnlp/sentiment140`.*

# G Details of Aggregation Rules

**FedAvg [24].** In FedAvg, a client aggregates the received shared models by calculating their average.

**Median [36].** Each client in Median obtains the post-aggregation local model via taking the coordinate-wise median of shared models.

**Trimmed-mean [36].** Trimmed-mean is also a coordinate-wise method that calculates the trimmed mean of the shared models in each dimension. Specifically, for dimension $k$, each client first removes the largest $c$ and smallest $c$ values, then computes the average of the remaining elements. In our experiments, we set $c = m$, where $m$ is the number of selfish clients, giving advantages to this aggregation rule.

Table 6: LSTM architecture for Sent140.

| Layer | Configuration |
|---|---|
| Embedding | Word embedding: GloVe [26] |
| | Embedding dimension: 50 |
| LSTM | Input size: 50 |
| | Hidden size: 100 |
| | Number of layers: 2 |
| Fully Connected | Input features: 100×2 |
| | Output features: 128 |
| Fully Connected | Input features: 128 |
| | Output features: 2 |

Table 7: Default DFL parameter settings.

| Parameter | CIFAR-10 | FEMNIST | Sent140 |
|---|---|---|---|
| # clients | | 20 | |
| # selfish clients | | 6 | |
| # local training epochs | | 3 | |
| Learning rate | | $1 \times 10^{-4}$ | |
| Optimizer | | Adam (weight decay=$5 \times 10^{-4}$) | |
| # global training rounds | 600 | 600 | 1,000 |
| Batch size | 128 | 256 | 256 |

**Krum [3].** In Krum aggregation rule, each client outputs one shared model that has the smallest sum of Euclidean distances to its closest $n - 2$ neighbors.

**FLTrust [5].** In FLTrust, client $i$ computes a trust score for each received shared model in each global training round, where $0 \leq i \leq n+m-1$. A shared model has a lower trust score if it deviates more from the pre-aggregation local model of client $i$. After that, client $i$ computes the weighted average of the received shared models, where the weights are determined by their respective trust scores. The higher the trust score, the larger the weight.

**FLDetector [37].** In FLDetector, client $i$ predicts a client's shared model updates based on its historical information in each training round and identifies a client as selfish when the predicted model updates consistently deviate from the shared model updates calculated by the received shared model across multiple training rounds. Then client $i$ employs Median to aggregate the predicted non-selfish clients' shared models.

**RFA [27].** The RFA algorithm employs a robust aggregation oracle based on the geometric median. It specifically uses the smoothed Weiszfeld algorithm to iteratively compute weights for aggregating the shared models. We follow the implementation as in Pillutla et al. [27].

**FLAME [25].** FLAME uses a clustering-based method to detect and eliminate bad shared models. Moreover, it uses dynamic weight-clipping and noise injection to further reduce the impact of bad shared models from selfish clients. We follow the implementation as in Nguyen et al. [25].

## H   Simulating Non-iid Setting in DFL

In DFL, the clients' local training data are typically not independent and identically distributed (Non-IID). We randomly split all clients into 10 groups, and then use a probability to assign a training example with label $y$ to the clients in one of these 10 groups. Specifically, the training example with label $y$ is assigned to clients in group $y$ with a probability of $\rho$, and to clients in any other groups with the same probability of $\frac{1-\rho}{9}$, where $\rho \in [0.1, 1.0]$. Within the same group, the training example is uniformly distributed among the clients. The parameter $\rho$ controls the degree of Non-IID. The local training data are IID distributed when $\rho = 0.1$, otherwise the training data are Non-IID. A larger value of $\rho$ implies a higher degree of Non-IID.

## I   Attacking FLAME

Recall that FLAME first clusters models based on cosine distances between them and selects the largest cluster with no fewer than $\frac{n+m}{2} + 1$ models. To encourage the clustering algorithm to mis-classify as many non-selfish clients as possible as selfish and subsequently remove them, we select a subset of non-selfish clients. We then craft shared models of selfish clients such that the crafted models are close to the shared models of these selected non-selfish clients, while keeping them distinctly different from the models of the remaining non-selfish clients. Specifically, when client $i$ employs the FLAME aggregation rule, we first identify $\frac{n-m}{2}$ shared models of non-selfish clients with the smallest cosine distance to $\boldsymbol{w}_i$, denoted as $\boldsymbol{w}_{(r_1,i)}, ...., \boldsymbol{w}_{(r_{(n-m)/2},i)}$. Then, we craft the shared model of selfish clients as follows:

$$
\boldsymbol{w}_{(n,i)} = \cdots = \boldsymbol{w}_{(n+m-1,i)} =
$$
$$
\frac{1}{\frac{n-m}{2} + \alpha - \beta n}\Big( \sum_{h=1}^{(n-m)/2} \boldsymbol{w}_{(r_h,i)} + \alpha\boldsymbol{w}_i - \beta \sum_{j=0}^{n-1} \boldsymbol{w}_{(j,i)}\Big), \tag{30}
$$

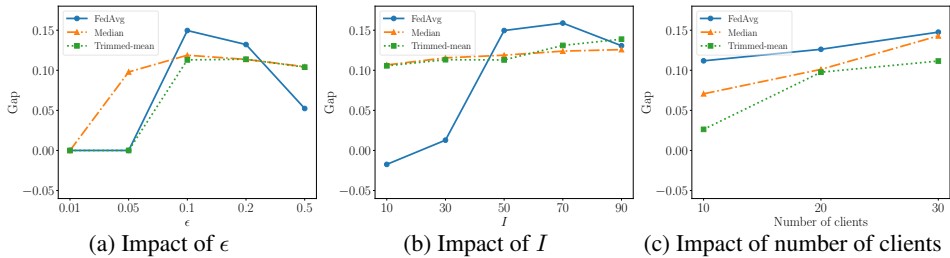

(a) Impact of $\epsilon$        (b) Impact of $I$        (c) Impact of number of clients

Figure 6: Impact of $\epsilon$, $I$, and total number of clients on Gap of SelfishAttack.

Table 8: Results when non-selfish clients and selfish clients use different aggregation rules. Each column represents the aggregation rule used by selfish clients, and each row denotes the rule used by non-selfish clients.

| Aggregation Rule | FedAvg (selfish) | | | Median (selfish) | | | Trimmed-mean (selfish) | | |
|---|---|---|---|---|---|---|---|---|---|
| | MTAS | MTANS | Gap | MTAS | MTANS | Gap | MTAS | MTANS | Gap |
| FedAvg | 0.475 | 0.326 | **0.150** | 0.478 | 0.326 | **0.152** | 0.470 | 0.326 | **0.144** |
| Median | 0.574 | 0.425 | **0.150** | 0.543 | 0.425 | 0.119 | 0.301 | 0.425 | −0.123 |
| Trimmed-mean | 0.597 | 0.484 | 0.113 | 0.596 | 0.484 | 0.113 | 0.597 | 0.484 | 0.113 |

where $\alpha > 0$ and $\beta > 0$ are some constants. In our experiments, we set $\alpha = 5$ and $\beta = 0.01$. In Equation 30, the first term aims to bring the crafted models closer to the selected $\frac{n-m}{2}$ shared models; the second term satisfies our utility goal, which is to maintain the proximity of client $i$'s post-aggregation local model to its pre-aggregation local model; and the third term pushes the crafted models away from the unselected models, which satisfies our competitive advantage goal. The selfish clients use the Median aggregation rule to aggregate received models from all clients.

It is noteworthy that although we do not devise specific attack strategies for FLAME's noise injection, the attack outlined in Equation 30 still enables selfish clients to gain a significant competitive advantage.

## J  Other Ablation Studies

**Impact of $\epsilon$ and $I$:** Figs. 6a and 6b show the impact of $\epsilon$ and $I$, respectively. The larger the $\epsilon$ and the smaller the $I$, the earlier the selfish clients start to attack. We observe that our SelfishAttack is more sensitive to these two parameters when attacking FedAvg compared to Median and Trimmed-mean. From Fig. 6a, we observe that when $\epsilon$ does not exceed 0.05, our attack yields a low Gap, especially when the aggregation rule is FedAvg or Trimmed-mean, where the Gap is 0. In fact, in this case, the selfish clients do not start to attack until the end of the training process. When $\epsilon$ ranges from 0.1 to 0.2, our attack exhibits relatively stable performance. When $\epsilon$ is 0.5 and the aggregation rule is FedAvg, selfish clients initiate the attack before the local models have almost converged, resulting in a decline in the attack effectiveness. We also observe from Fig. 6b that when attacking Median and Trimmed-mean, the performance of SelfishAttack is almost not affected by $I$, and the Gap obtained by our SelfishAttack is greater than $10\%$ in all cases. In contrast, when attacking FedAvg, the value of $I$ has a notable impact on Gap: when $I$ is not greater than 30, the Gap is close to 0; and when $I$ is not less than 50, the Gap is larger than $13\%$.

**Impact of total number of clients:** To explore the impact of the number of clients on SelfishAttack, we first divide the training data of CIFAR-10 dataset among 30 clients, and then randomly select subsets of 10, 20, and 30 clients from them while maintaining a fixed fraction of selfish clients at $30\%$. Fig. 6c shows the impact of the total number of clients on SelfishAttack. We observe that as the number of clients increases, our attack is more effective. For instance, when the aggregation rule is Median, increasing the number of clients from 20 to 30 results in a $4.2\%$ improvement in Gap. This may be because that although the fraction of selfish clients remains unchanged, the absolute number of selfish clients increases, thereby making the attack more effective.

**Impact of aggregation rule used by selfish clients:** Selfish clients can use different aggregation rules from non-selfish clients. Table 8 presents the results of SelfishAttack under varying combinations of aggregation rules used by selfish and non-selfish clients. For each aggregation rule used by

Table 9: Performance when selfish clients use previous round models from non-selfish clients.

| Aggregation Rule | MTAS | MTANS | Gap |
|---|---|---|---|
| FedAvg | 0.475 | 0.327 | 0.148 |
| Median | 0.545 | 0.427 | 0.118 |
| Trimmed-mean | 0.594 | 0.483 | 0.111 |

non-selfish clients, we apply the corresponding version of SelfishAttack. For instance, when non-selfish clients use Trimmed-mean, selfish clients craft their shared models using the Trimmed-mean based version of SelfishAttack. We observe that the Gap exceeds $11\%$ in most cases, with only one exception: when non-selfish clients use Median and selfish clients use Trimmed-mean. Overall, regardless of the aggregation rule employed by non-selfish clients, it is consistently a better choice for selfish clients to use FedAvg.

## K   Discussion and Limitations

**Utility preserving of SelfishAttack:**  SelfishAttack's utility goal requires selfish clients to learn more accurate local models than they would by training only with other selfish clients. According to Table 1, SelfishAttack's MTAS usually surpasses that of Two Coalitions, achieving this goal. Although sometimes selfish clients' model performance declines, we argue that this decrease is acceptable for two reasons. First, while MTAS shows a noticeable drop in some cases (e.g., FedAvg and Median on CIFAR-10) compared to No attack, the difference is usually within 6%. This drop could be attributed to the high degree of non-iid data in our CIFAR-10 setup. Notably, in some cases (e.g., FedAvg and Trimmed-mean on Sent140), SelfishAttack's MTAS even exceeds No attack. Second, this performance drop is compensated by a greater competitive advantage, as the Gap is consistently larger than the MTAS decline, meaning selfish clients sacrifice some model performance for more competitive gain.

**Defending against SelfishAttack:**  Byzantine-robust aggregation rules are defenses against "outlier" shared models, which can be applied to defend against SelfishAttack. However, we theoretically show that Median and Trimmed-mean cannot defend against our SelfishAttack, as the selfish clients can derive the optimal shared models to minimize a weighted sum of the two loss terms that quantify the two attack goals respectively. Moreover, we empirically show that other more advanced aggregation rules such as Krum, FLTrust, FLDetector, RFA, and FLAME cannot defend against SelfishAttack as well. It is an interesting future work to explore new defense mechanisms to defend against SelfishAttack. In SelfishAttack, a selfish client sends different shared models to different non-selfish clients. Therefore, one possible direction is to explore cryptographic techniques to enforce that a selfish client sends the same shared model to all clients. We expect such defense can reduce the attack effectiveness, though may not completely eliminate the attack since a selfish client can still send the same carefully crafted shared model to all non-selfish clients. Another direction for defense is to use trusted execution environment, e.g., NVIDIA H100 GPU. In particular, the local training and model sharing of each client is performed in a trusted execution environment, whose remote attestation capability enables a non-selfish client to verify that the shared models from other clients are genuine.

**Approximating with previous round models:**  A potential concern of our formulation is the information asymmetry in the model-sharing process. Specifically, the algorithm assumes that at each communication round, selfish clients can first obtain the shared models from all non-selfish clients before sending out their own models, which may be less realistic in practical decentralized or asynchronous federated learning systems. To examine whether this asymmetry affects the attack's effectiveness, we conduct an additional experiment where each selfish client instead uses the models received from each non-selfish client in the previous round as estimates of their current round models when constructing its selfish update. Table 9 reports the results on CIFAR-10. The results are very close to those obtained under the original assumption where selfish clients access the current round models. This indicates that the attack remains effective even when selfish clients rely on approximate, delayed information, suggesting that the method is robust to communication asymmetry and applicable to more realistic settings.

**Partial client participation:**  In practical FL systems, only a subset of clients may be active in each communication round due to resource or connectivity constraints. To examine the impact of such

Table 10: Results when only 50% of clients participate in each communication round.

| Aggregation Rule | MTAS | MTANS | Gap |
|---|---|---|---|
| FedAvg | 0.502 | 0.353 | 0.149 |
| Median | 0.583 | 0.465 | 0.118 |
| Trimmed-mean | 0.596 | 0.506 | 0.090 |

Table 11: Results under a partially connected client graph, where each client connects to 10 others.

| Aggregation Rule | MTAS | MTANS | Gap |
|---|---|---|---|
| FedAvg | 0.602 | 0.569 | 0.073 |
| Median | 0.505 | 0.388 | 0.117 |
| Trimmed-mean | 0.589 | 0.503 | 0.086 |

partial participation, we conduct an experiment where only 50% of the clients participate in each round, while keeping all other parameters unchanged. As shown in Table 10, our method continues to exhibit a strong competitive advantage under this setting. We further observe that using only half of the clients delays the manifestation of selfish behavior by about 20–30 rounds compared to using all clients, likely due to the slower convergence of the system when fewer participants are involved.

**Partially connected topology:** We further examine the robustness of SelfishAttack under a decentralized setting where the client network is not fully connected. Specifically, we randomly generate a communication graph in which each client connects to only 10 other clients (out of 20 in total). In this configuration, selfish clients do not have access to the full connectivity structure of the system. That is, they are unaware of which non-selfish clients are interconnected, and thus cannot directly apply the attack strategies proposed for the fully connected case. To approximate this scenario, we design an attack based on the local models of all non-selfish clients that are accessible to each selfish client, and apply the same attack strategy originally designed for FedAvg to all three aggregation rules (FedAvg, Median, and Trimmed-mean). As shown in Table 11, SelfishAttack still achieves a clear competitive advantage across all settings, indicating that the attack strategy generalizes effectively even when the system connectivity is partial.

## L   Broader Impact

Our work reveals a new type of insider threat in DFL systems, where a subset of selfish clients can manipulate the training process to gain a competitive advantage without disrupting overall model performance. By exposing this vulnerability, our research highlights the risks of trusting all participants in collaborative learning without strong assumptions or safeguards.

We believe this work will have a positive societal impact by raising awareness of subtle collusion threats in DFL and encouraging the development of more robust aggregation rules and detection mechanisms. At the same time, we acknowledge that malicious actors could potentially misuse our insights to harm collaborative learning systems in sensitive domains, such as healthcare or finance. To mitigate such risks, in Appendix K, we also discuss possible defenses against SelfishAttack. By highlighting both the threat and defense directions, we aim to facilitate the development of more secure and trustworthy decentralized learning systems.

