# OpenReview forum: "Competitive Advantage Attacks to Decentralized Federated Learning"
_NeurIPS.cc/2025/Conference — NeurIPS 2025 poster_

### Official Review · Reviewer_3xSU · 2025-07-01

**Clarity:** 3
**Significance:** 3
**Originality:** 4
**Rating:** 5
**Confidence:** 4

**Summary:**

This paper introduces SelfishAttack, a novel family of attacks targeting decentralized federated learning (DFL) systems. Selfish clients collude to manipulate shared models, aiming to achieve both a utility goal (improving their own model accuracy beyond standalone training) and a competitive-advantage goal (outperforming non-selfish clients).

**Questions:**

See above

**Ethical Concerns:**

["NO or VERY MINOR ethics concerns only"]

**Limitations:**

yes

**Quality:**

3

**Strengths And Weaknesses:**

Overall, this paper's idea is novel and interesting.

Strength:
1. This paper presents a novel approach with a sound and insightful underlying intuition, contributing meaningfully to the field of federated learning.;
2. The manuscript is well-written, clearly structured, and easy to follow.
3. The attack’s optimality is proven for FedAvg, Median, and Trimmed-mean, providing a solid foundation for its effectiveness.

Weakness:
1. The proposed attack can also be interpreted as an attempt to degrade the utility of other participants. Therefore, the paper should explicitly discuss the distinction between its approach and existing Byzantine or collusion-based attacks, such as those presented in [1, 2]. A clearer comparison would help situate the contribution within the broader landscape of adversarial strategies in federated learning.

[1] Fang M, Cao X, Jia J, et al. Local model poisoning attacks to {Byzantine-Robust} federated learning[C]//29th USENIX security symposium (USENIX Security 20). 2020: 1605-1622.

[2] Gu H, Fan L, Tang X, et al. Fedcut: a spectral analysis framework for reliable detection of byzantine colluders[J]. IEEE Transactions on Pattern Analysis and Machine Intelligence, 2024.

2. Given that the advantage of the proposed method is closely tied to fairness issues in federated learning, it is important that the paper includes a discussion of related work on fairness-aware FL algorithms. This would help clarify how the proposed method impacts or aligns with fairness goals in collaborative learning settings.

3. Current Byzantine-robust rules (e.g., Median, Trimmed-mean) cannot fully mitigate the attack, necessitating new defense mechanisms.

---

> ### Author Rebuttal · Authors · 2025-07-30
>
> We sincerely appreciate your recognition of our work and your valuable feedback. Please find our responses below.
>
> W1: **A clearer comparison to existing attacks**: Thank you for your positive feedback on our work. We discuss the "Comparison with poisoning attacks" of SelfishAttack in the Threat Model section. Overall, the differences between SelfishAttack and existing attacks are:
>
> 1). Adversary type: Byzantine or collusion-based attacks are typically launched by external adversaries or internal clients, while SelfishAttack is conducted by internal, coordinated clients within the system.
>
> 2). Objective: Byzantine or collusion-based attacks aim to indiscriminately degrade overall model performance, often harming even the compromised clients. SelfishAttack, on the other hand, aims to degrade non-selfish clients’ performance while improving or preserving the performance of selfish clients, thereby gaining a competitive advantage.
>
> W2: **Discussion of related work on fairness-aware FL**: We will consider incorporating some related works about fairness into our paper, such as [1, 2, 3].
>
> [1] Mohri, Mehryar, Gary Sivek, and Ananda Theertha Suresh. "Agnostic federated learning." International conference on machine learning. PMLR, 2019.
>
> [2] Shi, Yuxin, Han Yu, and Cyril Leung. "Towards fairness-aware federated learning." IEEE Transactions on Neural Networks and Learning Systems 35.9 (2023): 11922-11938.
>
> [3] Ezzeldin, Yahya H., et al. "Fairfed: Enabling group fairness in federated learning." Proceedings of the AAAI conference on artificial intelligence. Vol. 37. No. 6. 2023.
>
> W3: **New defenses**: We discussed potential defenses against SelfishAttack in Appendix K. For example, one possible direction is to explore cryptographic techniques to enforce that a selfish client sends the same shared model to all clients.

---

> > ### Comment · Reviewer_3xSU · 2025-08-04
> > **Reply to the authors**
> >
> > Thanks for your response, my concerns are addressed, thus, I hold my score at 5.

---

> > > ### Author Response · Authors · 2025-08-04
> > >
> > > Thank you for your response. We’re glad to hear that your concerns have been addressed. We truly appreciate your recognition of our work and your thoughtful review.

---

### Official Review · Reviewer_F5dV · 2025-07-03

**Clarity:** 3
**Significance:** 2
**Originality:** 3
**Rating:** 5
**Confidence:** 4

**Summary:**

This paper introduces SelfishAttack which is a new family of attacks in decentralized FL where a subset of clients are selfish and collude to improve their own local model performance with respect to the non-selfish clients (competitive advantage goal) without disrupting overall training (utility goal). The attack is formulated as an optimization problem to achieve the dual objectives. The paper also provides closed-form optimal attack solutions for popular aggregation rules - fedavg, trimmed mean, and median. Results on CIFAR-10, FEMNIST, and Sent140 show that the selfish clients reach upto 10-13% higher accuracy than non-selfish clients without getting filtered out by robust aggregators.

**Questions:**

The attack is started late as starting it early could poison the non-selfish nodes from the beginning and utility goal would not be reached. When is the training process stopped? It feels like early stopping would result in a dip in non-selfish accuracy (as desired) but continuing with the training could spread the attack infection, reducing the utility goal. Does the selfish accuracy keep on increasing even after starting the attack? How does the final selfish accuracy compare with the highest selfish/ non-selfish accuracy achieved over the training rounds. I ask this because it feels that if you just run the attack over the last few training rounds while not letting the selfish nodes learn from non-selfish models, you could easily achieve the competitive goal regardless of the actual attack method as long as the non-selfish models are being poisoned. Does the described attack reach a steady state in terms of the accuracy gap or does the gap keep increasing if the attack is continued?

In Table 2, why does selfish defense not improve selfish accuracy in selfish attack? The conclusion from Table 2 that the best strategy is to aggregate from all clients is based on just one experiment and cannot be generalized.

For the parameter settings described in Line 250 - approximately how many training rounds were benign and how many were attacked?

Why is \lambda set to 0 for FedAvg? How is competitive advantage still observed in Table 1 after setting it to 0?

Why does Trim attack have no impact on FedAvg training of Sent140?

What dataset is used in the results mentioned in Table 3?

How does the attacker control the post-aggregation models in Krum where only one model is chosen for aggregation, how is Krum fooled?

**Ethical Concerns:**

["NO or VERY MINOR ethics concerns only"]

**Final Justification:**

The biggest concern were two assumptions made in the original paper - malicious nodes having access to the full view of the system that is fully connected, and asymmetry in communication in the sense that the malicious nodes first receive the being updates, craft the attack and send the poisoned updates to the non-selfish nodes.

I had suggested the authors to consider partially connected communication graph and symmetry in communication by making use of the past updates from the non-selfish clients. They performed the experiments in the rebuttal and the results are positive.

I had further asked them to make changes in the threat model and the math accordingly with the changes in assumption, and they have done the hard work and made the changes, accepting that the new solution is a heuristic and not truly optimal in a more practical threat model, as suggested by me.

All concerns have hence been addressed. Therefore, I am increasing my score from 3 to 5.

**Limitations:**

Yes, they have, but this point is also worth mentioning if true -
The attack requires knowledge of current model updates across all the nodes to be successful. It would have been impressive if the selfish models can leverage the past updates of the non-selfish nodes to estimate and craft the attacked models. Working on the current update leads to a lag between receiving honest updates and sending back crafted models to a neighbor in the DFL setting increasing suspicion of foul play.

**Quality:**

2

**Strengths And Weaknesses:**

Strengths -

This novel attack does not poison the entire system, rather achieves competitive advantage over other nodes while maintaining utility.

Section 4 and Figure 2 are impressive and provide the methodology details very clearly, providing mathematical guarantees wherever possible.

Results in Table 1 are impressive showing unambiguous competitive advantage.

They compare both - individual and ‘two coalitions’ methods with the attack. The evaluation is sound. The utility goal is better confirmed with the ‘two coalitions’.

Weaknesses -

The greatest weakness in the attack method is that there are too many unsaid assumptions in the framework - it seems to be assumed that the communication graph is fully connected, or that the selfish clients have access to the gradients generated by all non-selfish clients. Line 24 says that all clients converge to identical models after each round - this is not true in a decentralized setting unless all nodes are connected and are using the exact same mixing weights as all other nodes. The fully connected graph seems to be an implicit assumption which may not hold in general decentralized settings (as can be even seen in Fig 1). The attack seems more appropriate in FL where a server has actual access to all client models, than in DFL. Knowledge of all non-selfish models makes the threat model very weak and thus the attack impact high.

In addition to the full view of the system and all client gradients, the attacker also needs to set the right \lambda value and the right ‘r’ value for good results. Together, these make the uncertainties too high, and without full knowledge of the system (which is unavailable in a DFL setting), the attack could not be as successful.

It appears that the selfish clients first receive the benign gradients from the non-selfish clients, process the received data to craft malicious models and then send the crafted models to them. This is not practical in a decentralized gossip setting. This delay is suspicious and would be indicative of an attack during the gossip stage itself.

Fig 4 shows that SelfishAttack outperforms other attacks but the attack baselines chosen are weak. The Fang attack mentioned in the FLTrust paper can be a stronger baseline to compare against.

---

> ### Author Rebuttal · Authors · 2025-07-30
>
> Thank you for acknowledging our work and for your insightful comments. Our responses are provided below.
>
> W1: **Unsaid assumptions**: We conduct an experiment under a not fully connected setting. Specifically, we randomly generate a graph structure where each client is connected to only 10 other clients (20 clients in total). In this case, selfish clients do not have access to the full connectivity structure of the DFL system, that is, they do not know which clients each non-selfish client is collaborating with. Therefore, it is difficult to directly apply the attack strategies proposed in the paper for FedAvg, Median, and Trimmed-mean in this setting. However, we can still construct an attack based on all the local models of non-selfish clients that are accessible to selfish clients. Therefore, we use the same attack strategy designed for FedAvg to attack FedAvg, Median, and Trimmed-mean. The results are as follows:
>
>
> |||
> |-|-|
> |FedAvg|0.602/0.569/0.073|
> |Median|0.505/0.388/0.117|
> |Trimmed-mean|0.589/0.503/0.086|
>
> These results demonstrate that SelfishAttack can still achieve a clear competitive advantage even in a not fully connected setting. Therefore, we believe that the attack strategy designed for FedAvg can be generalized to the not fully connected setting.
>
> W2: **Uncertainties**: For the different attack strategies proposed in our paper (FedAvg-based, Median-based, and Trimmed-mean-based), we use a fixed value of $\lambda$. Specifically, $\lambda$ is set to 0.0 for the FedAvg-based attack, and 0.5 or 1.0 for the Median- and Trimmed-mean-based attacks. This does not require full knowledge of the system and remains constant regardless of any other factors. Figure 3 is only intended to illustrate the motivation behind our choice of $\lambda$ for these three strategies. Table 3 presents results of attacks against other aggregation rules. In these experiments, we adopted the FedAvg-based attack, and thus $\lambda$ is fixed at 0.0 accordingly.
> As for r in Equation 8, it is computed directly and does not require manual selection. Therefore, our experimental parameters are fixed and do not rely on any specific knowledge of the dataset or DFL system.
>
> W3: **Delay is suspicious**: We experiment with using the models received from each non-selfish client in the previous round, as estimates of their current-round sent models, when computing the selfish model to be sent by selfish clients. The results (CIFAR-10, MTAS/MTANS/Gap) are as follows:
>
> |||
> |-|-|
> |FedAvg|0.475/0.327/0.148|
> |Median|0.545/0.427/0.118|
> |Trimmed-mean|0.594/0.483/0.111|
>
> These results are very close to those obtained using the actual current-round models. We will consider updating our algorithm to use the previous-round models to address this concern.
>
> W4: **Stronger Baseline**: In fact, the Fang attack is equivalent to the TrimAttack described in our paper. We further include its Krum-based variant, referred to as the Krum Attack. The results are as follows:
>
> ||KrumAttack|
> |-|-|
> |FedAvg|0.103/0.100/0.003|
> |Median|0.598/0.589/0.009|
> |Trimmed-mean|0.611/0.609/0.002|
>
> Q1: **The attack is started late…**: In fact, for CIFAR-10, the attack typically begins around round 200, and once initiated, selfish clients can achieve a clear competitive advantage within just 10 rounds. After that point, the average accuracy of both selfish and non-selfish clients increases slightly (by about 0.05), and the Gap tends to stabilize until training ends (at 600 rounds). Therefore, neither early stopping nor continuing training significantly affects the effectiveness of SelfishAttack. As long as the attack is launched after a certain round, selfish clients can consistently gain a stable competitive advantage.
>
> We test the effectiveness of SelfishAttack when launched during the final 50 rounds of training, while also preventing selfish clients from learning from non-selfish clients. The resulting Gaps are 0.132 for FedAvg, 0.116 for Median, and 0.110 for Trimmed-mean, which are relatively close to the results in Table 1. However, 'early stopping' can be an effective defense against such 'last-round attacks,' especially in settings where the model is continuously updated and the end of training is uncertain.
>
> Q2: **Selfish defense does not improve selfish accuracy**: Here, the 'Selfish' entry under the 'Info' column does not refer to a defense mechanism, but rather indicates whether the model used during aggregation by selfish clients comes from all clients or only from selfish clients. The reason why 'All' outperforms 'Selfish' is that selfish clients can still gain some useful information from the models of non-selfish clients. In addition, we add experiments where non-selfish clients use Median, selfish clients perform a Median-based attack, and use FedAvg to aggregate the selfish clients' local models. The results are as follows:
>
> |Info.|TrimAttack|GaussianAttack|SelfishAttack|
> |-|-|-|-|
> |All|0.524/0.513/0.011| 0.619/0.617/0.001|0.543/0.425/0.119|
> |Selfish|0.510/0.513/-0.003|0.510/0.617/-0.107|0.510/0.425/0.085|
>
> This further supports the conclusion that models from all clients should be used for aggregation, rather than relying solely on those from the selfish clients.
> Q3: **Number of benign and malicious training rounds**: Roughly, on CIFAR-10, the first 200 rounds are benign, and the later 400 rounds are attacked.
>
> Q4: **Why is \lambda set to 0 for FedAvg**: Because FedAvg is not a robust aggregation rule, using a large value of $\lambda$ would cause the model aggregated by non-selfish clients to deviate significantly from the model aggregated solely from all non-selfish clients' models. This would lead to poor performance for the non-selfish clients, preventing the selfish clients from benefiting from them. For the impact of different $\lambda$ values on performance, please refer to Figure 3.
>
> Q5: **Trim attack has no impact on Sent140**: This may be because Sent140 is a binary classification task, where the worst-case performance (i.e., random guessing) is 0.5. In addition, the TrimAttack also has some impact, causing the average accuracy of non-selfish clients to drop by around 0.08.
>
> Q6: **Dataset used in Table 3**: It’s CIFAR-10, which is our default setting.  We mentioned our default settings in line 251.
>
> Q7: **How krum is fooled**: Because Krum selects the model that has the smallest sum of Euclidean distances to its closest (n+m)−m−2 neighbors, and most selfish clients send models that are identical to the local model of the targeted non-selfish client, the distances between these models are nearly 0. As a result, Krum is highly likely to select the model sent by a selfish client.
>
> L1: **Leverage the past updates**: Thank you for your suggestion. We experiment with using the past models received from each non-selfish client. Please see the response to W3 for more details.

---

> > ### Comment · Reviewer_F5dV · 2025-08-01
> > **Good rebuttal, but core threat model concerns remain.**
> >
> > Thank you for all the clarification (W4, Q2, Q5-7) and presenting the new results with the suggested experiments.
> >
> > The new details in Q1 (dynamics of the training process) and Q4 (lambda for fedavg) make the work look stronger now, but I still have a few concerns.
> >
> > Now, if we have the selfish clients use past models and do not assume the graph is fully connected, does it invalidate the math (estimating w, calculating r, attack bounds), losing the optimality guarantee? How is r calculated with these changes? The algorithm will no longer be solving for the true optimal but for an approximation and becomes a heuristic, which is still powerful as per the new results shared.
> >
> > For a heuristic as an approximate solution, it would be ideal to evaluate it across multiple datasets, but i assume the results may still be good if it works well on CIFAR10. So a recommendation would be make revisions in the main paper with much more transparency about the threat model and optimality.

---

> ### Author Response · Authors · 2025-08-04
>
> Thank you for your positive feedback on our rebuttal!
>
> First, we redefine the threat model under the setting where the communication graph may not be fully connected.
>
> **DFL Settings**
>
> The communication graph may not be fully connected. This means each non-selfish client only sends and receives its local model to and from a subset of clients.
>
> **Attacker's Knowledge**
>
> Each selfish client receives the pre-aggregation local models from all non-selfish clients it is connected to. They also share their own pre-aggregation local models, as well as all local models they have received from their neighboring non-selfish clients, among themselves.
>
> We consider two scenarios:
>
> i) Selfish clients know the structure of the communication graph, i.e., they know which clients each non-selfish client is connected to;
>
> ii) Selfish clients do not know the structure of the communication graph.
>
> We then discuss how theoretical guarantees of the algorithm change when the attacker knows or does not know the graph structure, and when the attacker can only use local models received in the previous round to approximate the optimal solution.
>
> Let $A_ i$ denote the set of indices of all neighbors of client $i$. Suppose there are $n_ i$ non-selfish clients and $m_ i$ selfish clients in $A_ i$. Denote the model sent from client $i$ to $j$ in round $t$ as $w_ {(i,j)}^{(t)}$. After approximating using shared models from non-selfish clients in round $t-1$, our objective function is defined as:
> $$
> \min_ {\{w_ {(j,i)}^{(t)}\}_ {n\leq j\leq n+m-1}}\Vert w^{(t)}_ i- \hat{w}^{(t-1)}_ i \Vert^2 - \lambda \Vert w^{(t)}_ i -\tilde{w}^{(t-1)}_ i \Vert^2,
> $$
> where $w_ i^{(t)} = \text{Agg}(\{w_ {(h,i)}^{(t)}\}_ {h\in A_ i})$, $\tilde{w}_ i^{(t-1)}  = \text{Agg}(\{w_ {(h,i)}^{(t-1)}\}_ {0\leq h\leq n-1})$. The unconstrained minimizer now is $p_ i^{(t)}[k] = \frac{\hat{w}_ i^{(t-1)}[k] - \lambda \tilde{w}_ i^{(t-1)}[k]}{1-\lambda}$, and the optimal solution is computed using $p_ i^{(t)}$.
>
> We first consider the scenario in which selfish clients know the structure of the communication graph; then they can craft shared models similar to the methods described in the main paper. More specifically, we sort the $k$th-dimensional shared models from non-selfish clients to $i$ in descending order in round $t-1$ as ${q}_ {0i}^{(t-1)}[k] \ge \cdots \ge {q}_ {(n_ i-1)i}^{(t-1)}[k]$. Let all the indices of selfish clients in $A_ i$ be $j_ 0, \ldots, j_ {m_ i-1}$.
>
> For **FedAvg**, the bounds become $\overline{w}_ i^{(t)}[k] = \max_ {0\leq h\leq n-1} w_ {(h,i)}^{(t-1)}[k]$ and $\underline{w}_ i^{(t)}[k] = \min_ {0\leq h\leq n-1} w_ {(h,i)}^{(t-1)}[k]$, and shared models become:
> $$w_ {(j_ 1,i)}^{(t)}[k]=\cdots=w_ {(j_ {m_ i-1},i)}^{(t)}[k]=w_ i^* [k], \
> w_ {(j_ 0,i)}^{(t)}[k]=(n_ i+1)\cdot w_ i^* [k] - \sum_ {h=0}^{n_ i-1}q_ {hi}^{(t-1)}[k].$$
>
> For **Median**, the bounds become
> $$
> \overline{w}_ i^{(t)}[k] = \frac{1}{2}(q_ {\lfloor\frac{n_ i-m_ i-1}{2}\rfloor i}^{(t-1)}[k] + q_ {\lfloor\frac{n_ i-m_ i}{2}\rfloor i}^{(t-1)}[k]), \quad
> \underline{w}_ i^{(t)}[k] = \frac{1}{2}(q^{(t-1)}_ {\lfloor\frac{n_ i+m_ i-1}{2}\rfloor i}[k] + q^{(t-1)}_ {\lfloor\frac{n_ i+m_ i}{2}\rfloor i}[k]).
> $$
> and shared models become
> $$
> w_ {(j_ 1,i)}^{(t)}[k] = \cdots = w_ {(j_ {m_ i-1},i)}^{(t)}[k] = w_ i^* [k],
> $$
> $$
> w_ {(j_ 0,i)}^{(t)}[k] =
> \begin{cases}
> 2w_ i^* [k] - q_ {ui}^{(t-1)}[k], & \text{if } w_ i^* [k] > q_ {ui}[k], \\\\
> 2w_ i^* [k] - q_ {vi}^{(t-1)}[k], & \text{if } w_ i^* [k] < q_ {vi}[k], \\\\
> w_ i^* [k], & \text{otherwise}
> \end{cases}
> $$
>
> For **Trimmed-mean**, the bounds become
> $$
> \overline{w}_ i^{(t)}[k] = \frac{1}{n_ i - m_ i} \sum_ {h=0}^{n_ i - m_ i - 1} q_ {hi}^{(t-1)}[k], \quad
> \underline{w}_ i^{(t)}[k] = \frac{1}{n_ i - m_ i} \sum_ {h=m_ i}^{n_ i - 1} q_ {hi}^{(t-1)}[k].
> $$
> and shared models become
>
> Case I: $$
> w_ {(j,i)}^{(t)}[k]=
> \begin{cases}
> q^{(t-1)}_ {(n_ i-1)i}[k]-b, &  j\in\\{j_ {n_ i-r_ i-1},...,j_ {m_ i-1}\\}\\\\
> {\mathbf{c}}^{(t)}_ i[k], & j\in\\{j_ 0,...,j_ {n_ i-r_ i-1}\\}
> \end{cases}
> $$
> where $
> \mathbf{c}_ i^{(t)}[k] = \frac{(n_ i - m_ i)\cdot w_ i^* [k] - \sum_ {h = m_ i}^{r_ i - 1} q_ {hi}^{(t-1)}[k]}{n_ i - r_ i}$, $r_ i$ is the largest integer to ensure $n_ i-m_ i\leq r_ i\leq n_ i$ and $w^* _ i[k]\leq \frac{1}{n_ i-m_ i}((n_ i-r_ i)\cdot q^{(t-1)}_ {(m_ i-1)i}[k]+\sum_ {h=m_ i}^{r_ i-1}q^{(t-1)}_ {hi}[k])$
>
> Case II is similar.
>
> When the selfish clients do not know the graph structure, they cannot accurately obtain the bounds used in median and trimmed-mean, nor the shared models. Therefore, we adopt a heuristic approach, allowing them to launch the attack as if targeting FedAvg. Below are our experimental results under different datasets in the setting where the graph is not fully connected and past models are used.
>
> ||CIFAR-10|FEMNIST|Sent140|
> |-|-|-|-|
> |FedAvg|0.599/0.527/0.072|0.758/0.685/0.073|0.793/0.711/0.082|
> |Median|0.585/0.452/0.133|0.764/0.633/0.131|0.824/0.669/0.155|
> |Trimmed-mean|0.613/0.540/0.073|0.774/0.724/0.050|0.785/0.692/0.093|

---

> > ### Comment · Reviewer_F5dV · 2025-08-04
> > **Convincing response**
> >
> > Thank you for making the suggested changes. This makes the paper even stronger. The authors have addressed all of my concerns. I will update my score now.

---

> > > ### Author Response · Authors · 2025-08-04
> > >
> > > Thank you for your encouraging feedback. We’re happy to hear that our revisions addressed your concerns, and we appreciate your continued support.

---

### Official Review · Reviewer_iRFS · 2025-07-03

**Clarity:** 3
**Significance:** 2
**Originality:** 3
**Rating:** 4
**Confidence:** 3

**Summary:**

This paper introduces a novel attack in the decentralized federated learning paradigm, called SelfishAttack, in which a subset of selfish clients aim to gain a competitive advantage over the remaining non-selfish clients while maintaining the performance of their own trained models. The authors propose specific selfish attack algorithms tailored to different aggregation rules and demonstrate their effectiveness through numerical experiments.

**Questions:**

**Q1:**

If the selfish agents do not know the aggregation rule in advance, which selfish attack strategy should they adopt? Among the attack strategies designed for FedAvg, Median, and Trimmed-Mean, is there one that performs better on average when the aggregation rule is unknown?

---

**Q2:**

In each communication round, if only a random subset of agents is selected to participate in the model update, how does this affect the effectiveness of the SelfishAttack algorithms? How does it influence the optimal timing for initiating the attack?

---

**Q3:**

It seems to me that the aggregation rule proposed in [1] might be able to defend against the SelfishAttack. Specifically, in that work, the authors suggest that each agent $i$ assigns weights to received models based on their relative performance on the agent's local dataset compared to agent's own model. This could potentially help distinguish between models from selfish and non-selfish agents . It would be helpful if the authors could comment on this point.

[1] Zhang, Michael, et al. "Personalized Federated Learning with First Order Model Optimization." International Conference on Learning Representations.

**Q4:**

While I understand this may not be the main focus of the paper, could the authors comment on how the proposed method could potentially be generalized to scenarios where the network topology of DFL is not a fully connected graph?

**Ethical Concerns:**

["NO or VERY MINOR ethics concerns only"]

**Final Justification:**

I have read all the reviewers’ comments and the corresponding replies from the authors (including my own). The experiments presented in the paper, together with the additional experiments in the rebuttal period, suggest that the proposed Selfish Attacks can successfully attack many existing DFL algorithms. In my view, this will be of interest to the community. Accordingly, I recommend borderline acceptance.

**Limitations:**

Yes

**Quality:**

2

**Strengths And Weaknesses:**

### **Strengths**

1. The paper proposes a new type of attack in the setting of decentralized federated learning, which is both a reasonable threat to consider and, to the best of my knowledge, novel.

2. The proposed algorithms successfully achieve the goals of the selfish attack under various robust aggregation rules, demonstrating their effectiveness.

### **Weaknesses**

1. The proposed algorithm seems impractical in some respects. In particular, at each communication round, the selfish agents are assumed to first obtain the shared models from all the non-selfish agents before sharing their own models. This asymmetry—where selfish agents receive others’ models before sending theirs—may not be realistic in real-world decentralized systems.

2. The algorithm assumes that the selfish agents know in advance which aggregation rule is being used by the non-selfish agents. In practice, this information may not be available. It is also unclear what would happen if, for example, the attack designed for Median aggregation were applied when the non-selfish agents were actually using Trimmed-mean aggregation. While I saw that the authors mentioned this part in the future work, but I think Including experiments to evaluate such cases would strengthen the work.

3. The proposed algorithm assumes that all agents participate in every communication round. However, in practice, participation is often partial, with only a subset of agents active in each round. While it seems plausible that the algorithm could still be effective under partial participation, it would strengthen the paper to include experiments under this more realistic scenario. Such experiments could also shed light on how participation randomness impacts the effectiveness of the attack and the choice of hyperparameters, such as the timing of when to initiate the attack.

---

> ### Author Rebuttal · Authors · 2025-07-30
>
> We thank you for your constructive feedback and for recognizing the contributions of our work. Our point-by-point responses are as follows.
>
> W1: **Asymmetry**: Regarding the 'asymmetry' concern, we experiment with using the models received from each non-selfish client in the previous round, as estimates of their current-round sent models, when computing the selfish model to be sent by selfish clients. The results (CIFAR-10, MTAS/MTANS/Gap) are as follows:
>
> |||
> |-|-|
> |FedAvg|0.475/0.327/0.148|
> |Median|0.545/0.427/0.118|
> |Trimmed-mean|0.594/0.483/0.111|
>
> These results are very close to those obtained using the actual current-round models. We will consider updating our algorithm to use the previous-round models to address this concern.
>
> W2: **Aggregation rules are unknown**: We include several advanced aggregation rules (e.g., FLTrust, RFA, FLDetector) in Table 3. These experiments do not require prior knowledge of which aggregation rule the clients are using, as we apply a unified attack strategy rather than adapting it based on the specific aggregation rule. Of course, if the attacker knows in advance which aggregation rule the non-selfish clients are using, it is possible to launch more targeted and effective attacks (as we did for FedAvg, Median, and Trimmed-mean). We also attempt to attack Trimmed-mean using the strategy designed for Median as suggested, resulting in MTAS/MTANS/Gap of **0.583/0.512/0.071**, demonstrating a certain degree of transferability.
>
> W3: **A subset of agents active in each round**: We conduct an experiment where only 50% of the clients participate in each round, while keeping other parameters unchanged (e.g., $\lambda$). The results are as follows:
>
> |||
> |-|-|
> |FedAvg|0.502/0.353/0.149|
> |Median|0.583/0.465/0.118|
> |Trimed-mean|0.596/0.506/0.090|
>
> The results still demonstrate that our method achieves a strong competitive advantage. In addition, using only 50% of the clients delays the attack by 20-30 rounds compared to using all clients, which may be due to the slower convergence of the system with fewer participants.
>
> Q1: **Aggregation rules are unknown**: If the aggregation rules are not known in advance, using the attack strategies designed for FedAvg is the most effective approach. This is also consistent with our experiments on most other aggregation rules in Table 3. Under the same parameter settings, applying the FedAvg-based attack against Median and Trimmed-mean yields the following results:
>
> |||
> |-|-|
> |Median|0.571/0.464/0.107|
> |Trimmed-mean|0.579/0.495/0.084|
>
> Q2: **A subset of agents active in each round**: Please refer to the responses to W3 above.
>
> Q3: **Potential defense**: We evaluate the aggregation rule FedFomo proposed in [1]. Similar to how we attacked other aggregation rules in Table 3, we apply the attack strategy designed for FedAvg to FedFomo. In our experiments, each non-selfish client uses its entire training dataset to evaluate the models from other clients. The results are **0.474/0.323/0.151**. This outcome may be due to the fact that, under our attack on FedAvg, most selfish clients send models identical to the local models of the non-selfish clients. Since FedFomo assigns higher weights to models that are similar to a client’s own local model, the models from selfish clients receive relatively high weights. As a result, the final aggregated model essentially becomes the non-selfish client's own model. This explains why the attack results on FedFomo are very close to those on FedAvg.
>
> Q4:**Not fully connect graph**:  We conduct an experiment under a not fully connected setting. Specifically, we randomly generate a graph structure where each client is connected to only 10 other clients (20 clients in total). In this case, selfish clients do not have access to the full connectivity structure of the DFL system, that is, they do not know which clients each non-selfish client is collaborating with. Therefore, it is difficult to directly apply the attack strategies proposed in the paper for FedAvg, Median, and Trimmed-mean in this setting. However, we can still construct an attack based on all the local models of non-selfish clients that are accessible to selfish clients. Therefore, we use the same attack strategy designed for FedAvg to attack FedAvg, Median, and Trimmed-mean. The results are as follows:
>
>
> |||
> |-|-|
> |FedAvg|0.602/0.569/0.073|
> |Median|0.505/0.388/0.117|
> |Trimmed-mean|0.589/0.503/0.086|
>
> These results demonstrate that SelfishAttack can still achieve a clear competitive advantage even in a not fully connected setting. Therefore, we believe that the attack strategy designed for FedAvg can be generalized to the not fully connected setting.

---

> > ### Comment · Reviewer_iRFS · 2025-08-03
> > **Reply to the authors**
> >
> > I would like to thank the authors for providing detailed explanations that address my questions and concerns. In my view, all of the points I raised have been satisfactorily addressed. The added experiment are great. I reconmmand the authors to incorporate them into the paper during the revision. Accordingly, I have raised my score.

---

> > > ### Author Response · Authors · 2025-08-04
> > >
> > > Thank you again for your thoughtful and encouraging review. We’re glad that our responses addressed your concerns, and we sincerely appreciate your decision to raise your score.

---

### Official Review · Reviewer_2QD8 · 2025-07-03

**Clarity:** 2
**Significance:** 2
**Originality:** 3
**Rating:** 5
**Confidence:** 3

**Summary:**

SelfishAttack is a collaborative attack in which a group of malicious clients send corrupt updates to (i) enhance their own final models beyond what they could achieve independently and (ii) widen the performance gap. The attack uses a competitive-advantage term to push towards suboptimal regions. Closed-form solutions are derived for standard aggregation rules like FedAvg, Median, and Trimmed-mean. Additionally, a transferable FedAvg-based attack strategy proves effective against more sophisticated defenses, including Krum, FLTrust, FLDetector, RFA, and FLAME. Experiments on CIFAR-10, FEMNIST, and Sent140.

**Questions:**

1.) Is the proposed method permutation invariant? Karimireddi et.al showed that permutation methods cannot be strongly robust from byzantine attacks.
2. Shouldn't attacks be benchmarked against defenses instead of with just other attacks?
3. Can the problem be written as a game? That is, if defenders start randomizing aggregation weights or tracking per‐round variance or use an advanced defense, could the colluding clients dynamically adjust their optimization to preserve their advantage? Modeling this game and reporting the performance trade-offs would be of greater practical usage.

**Ethical Concerns:**

["NO or VERY MINOR ethics concerns only"]

**Final Justification:**

Based on the concerns raised by rest of the reviewers, the strong rebuttals, and backing up of the threat model, flexibility to connectedness level of graph, I revised and raised my score.

**Limitations:**

Weaker performance in non-iid data is a prominent limitation of this approach.
The method is not effective during early rounds of local training.

**Paper Formatting Concerns:**

References 3 and 4 are exactly the same. Please fix it.

**Quality:**

2

**Strengths And Weaknesses:**

Detailed derivations for FedAvg, Median, and Trimmed-mean provide a first-principles based application.
The per-coordinate re-parameterization yields clean closed-form optimal updates. The ablations are in good detail.

Weaknesses:
i) The experiments compared with simple robust aggregation such as median and trimmed mean. There has been a lot of progress beyond this with say the CenteredClip Aggregation (Karimireddi et.al), Karimireddy, S. P., He, L., and Jaggi, M. (2021b). Learning from history for byzantine robust optimization. In International Conference on Machine Learning. It is good to compare against it.
ii) That said, the number of clients considered is small and the datasets used are fairly simple.

Similarly on the attack side, do contrast the setting considered from those of label flipping attacks, sign flipping attacks, inner-product manipulation attacks and alie attacks that have been quite popular.

---

> ### Author Rebuttal · Authors · 2025-07-30
>
> Thank you for your comments and feedback. Please find our responses below.
>
> W1: **Simple robust aggregation**: In Table 3, we present results for other aggregation rules. We also conduct experiments on complex CenteredClip aggregation rule. Similar to the attacks against other aggregation rules in Table 3, we apply the attack originally designed for FedAvg to CenteredClip. The corresponding MTAS/MTANS/Gap are **0.527/0.437/0.090**, respectively. These results indicate that our attack successfully achieves both attack goals even under the CenteredClip defense.
>
> W2: **Few clients**: To further evaluate the effectiveness of our approach, we conduct experiments on CIFAR-10 with 100 clients. The corresponding MTAS/MTANS/Gap results are:
>
> |||
> |-|-|
> |FedAvg|0.238/0.167/0.071|
> |Median|0.394/0.239/0.155|
> |Trimmed-mean|0.425/0.301/0.124|
>
> W3: **More attacks**: We further add label flipping and sign flipping attacks to our baseline. The results are:
>
> ||Label Flipping|Sign Flipping|
> |-|-|-|
> |FedAvg|0.597/0.553/0.044 | 0.565/0.563/0.002 |
> |Median| 0.616/0.601/0.015 | 0.597/0.589/0.008 |
> |Trimmed-Mean| 0.610/0.591/0.019 | 0.585/0.579/0.006 |
>
> These results indicate that the two attacks fail to obtain a significant competitive advantage.
>
> Q1: **Permutation invariant**: Permutation invariance is an important property for FL aggregation rules. However, since our method is an attack in the DFL setting, the concept of 'permutation invariance' may not be applicable to our method.
>
> Q2: **Attacks should be benchmarked against defenses**:  In addition to comparing the performance of our attack with other attacks, we also benchmarked against defenses. Tables 1 and 3 report the results of SelfishAttack against various robust aggregation rules. Moreover, comparing our attack with other baseline attacks better highlights its effectiveness, which is also a common practice in many attack papers (e.g., [1]).
>
> [1] Fang et al. "Local model poisoning attacks to Byzantine-Robust federated learning." In USENIX security symposium 2020.
>
> Q3: **Modeling as a game**: This is a very interesting suggestion. In Table 3, we have already included several advanced defenses, such as FLTrust, FLAME, and RFA, which modify aggregation weights, as well as FLDetector, which tracks per-round variance. To further verify that attackers can still be effective even when defenders adopt random aggregation rules, we conduct the following experiment: in each round, each non-selfish client randomly selects an aggregation rule from those listed in Tables 1 and 3, while the selfish clients consistently apply the attack strategy designed for FedAvg. The results are **0.573/0.480/0.093**. While an attacker could potentially adapt its strategy based on the specific aggregation rule used by the defender, it needs the selfish clients to know the exact aggregation rules used by non-selfish clients, which is impractical.  Therefore, we believe modeling games goes beyond the scope of the current paper and would be better positioned as a new line of attack. We believe it is a promising future work.
>
> L1: **Weaker performance in non-iid**: We clarify that all our results, except for the iid case in Figure 4 (i.e., degree of non-iid=0.1), are conducted under non-iid scenarios. We will make this setting more clear.
>
> L2: **Not effective during early rounds**: We argue that in early rounds, all clients have poorly performing local models, and a more meaningful attack goal is to prevent them from obtaining high-performance models later on. Our algorithm includes a component that determines when to start the attack. In practice, the attack typically begins between 1/6 and 1/3 of the total training rounds, meaning that selfish clients can gain a significant competitive advantage for the majority of the training process.

---

> > ### Comment · Reviewer_2QD8 · 2025-08-04
> > **thank you for the responses**
> >
> > Thank you for the responses and the updated results on W1 and W2 . Based on the answers to Q2, Q3, and L2, and the rest of reviewers concerns, and positives, I hold my score at 4.

---

> > > ### Author Response · Authors · 2025-08-04
> > >
> > > Thank you for your thoughtful review and for taking the time to consider our responses and updated results. We appreciate your feedback and your continued engagement with our work.

---

### Decision · Program_Chairs · 2025-09-17

**Decision:**

Accept (poster)

**Comment:**

This paper received the following ratings: Accept, Borderline accept, Accept, Accept. The authors present SelfishAttack, an adversarial strategy in decentralized federated learning, where a group of selfish clients collude to improve their own model performance while degrading others non-selfish clients. The attack is mathematically optimized for various aggregation rules (FedAvg, Median, Trimmed-mean) and shows strong results across multiple datasets. This new attack does not poison the entire system, rather achieves competitive advantage over other nodes while maintaining utility, which makes it harder to detect. Most reviewers find the proposed approach novel, with a sound and insightful underlying intuition. Solid mathematical justifications have been provided wherever possible.  AC recommends accepting the paper.